# Eradicating the large white butterfly from New Zealand eliminates a threat to endemic Brassicaceae

Craig B. Phillips [1,2]*, Kerry Brown[3], Chris Green[3], Richard Toft[4], Graham Walker[2,5], Keith Broome[3]

1 Biocontrol and Biosecurity Group, AgResearch, Lincoln, New Zealand, 2 Better Border Biosecurity research collaboration, www.b3nz.org, Aotearoa, New Zealand, 3 Department of Conservation, Wellington, New Zealand, 4 Entecol Ltd, Nelson, New Zealand, 5 Plant & Food Research, Auckland, New Zealand

* craig.phillips@agresearch.co.nz

**Data Availability Statement:** All relevant data are within the manuscript and its Supporting Information files (S1 Data). The data have been aggregated spatially and/or temporally to protect

## Abstract

In May 2010 the large white butterfly, *Pieris brassicae* L. (Lepidoptera: Pieridae), was discovered to have established in New Zealand. It is a Palearctic species that—due to its wide host plant range within the Brassicaceae—was regarded as a risk to New Zealand's native brassicas. New Zealand has 83 native species of Brassicaceae including 81 that are endemic, and many are threatened by both habitat loss and herbivory by other organisms. Initially a program was implemented to slow its spread, then an eradication attempt commenced in November 2012. The *P. brassicae* population was distributed over an area of approximately 100 km$^2$ primarily in urban residential gardens. The eradication attempt involved promoting public engagement and reports of sightings, including offering a bounty for a two week period, systematically searching gardens for *P. brassicae* and its host plants, removing host plants, ground-based spraying of insecticide to kill eggs and larvae, searching for pupae, capturing adults with nets, and augmenting natural enemy populations. The attempt was supported by research that helped to progressively refine the eradication strategy and evaluate its performance. The last New Zealand detection of *P. brassicae* occurred on 16 December 2014, the eradication program ceased on 4 June 2016 and *P. brassicae* was officially declared eradicated from New Zealand on 22 November 2016, 6.5 years after it was first detected and 4 years after the eradication attempt commenced. This is the first species of butterfly ever to have been eradicated worldwide.

## Introduction

Unintentional introductions of nonnative species, including arthropods, are contributing to declining global biodiversity [1–3]. Eradicating destructive nonnative species is challenging, but when successful can provide substantial benefits [4,5]. The first organised attempt to eradicate a nonnative arthropod probably began in 1890 against the gypsy moth, *Lymantria dispar*, in the USA [6]. Subsequently over 1200 programs in about 100 countries have attempted to

the identities of the numerous individual properties that were sampled during this work. The data provided in S1 Data are sufficient to support the results and conclusions of the work presented in the manuscript.

**Funding:** Details of all costs are publicly available on-line in the Department of Conservation's 2015-16 Pieris brassicae eradication program annual report: www.doc.govt.nz/about-us/sciencepublications/conservation-publications/threats-and-impacts/animal-pests/pieris-brassicae-greatwhite-butterfly-eradication-annual-report/ Operational aspects of the eradication program were funded by the New Zealand Department of Conservation (DOC; www.doc.govt.nz). Vegetables New Zealand (www.freshvegetables.co.nz) contributed some funds to DOC to support operational aspects of the eradication program. DOC provided support in the form of salaries for authors K. Brown, CG and K. Broome, but did not have any additional role in the study design, data collection and analysis, decision to publish, or preparation of the manuscript. The specific roles of these authors are articulated in the 'author contributions' section. Vegetables New Zealand did not have any additional role in the study design, data collection and analysis, decision to publish, or preparation of the manuscript. RT is a the Managing Director of a commercial company EntEcol Ltd (www.entecol.co.nz) which provides technical entomological services to New Zealand clients. In the eradication program, EntEcol Ltd was contracted by DOC for RT to provide services including contributing to the TAG, preparing documents, identifying specimens, helping to develop the visual lure, and evaluating P. brassicae parasitism rates. EntEcol Ltd provided support in the form of a salary for author RT, but did not have any additional role in the study design, data collection and analysis, decision to publish, or preparation of the manuscript. The specific roles of this author is articulated in the 'author contributions' section. The New Zealand government research institutes AgResearch (www.agresearch.co.nz) and Plant and Food Research (www.plantandfood.co.nz) are partners in a New Zealand research collaboration called Better Border Biosecurity (www.b3nz.org). The collaboration aims to help reduce the rate at which non-native insects, weeds and diseases that could harm valued New Zealand plants are becoming established in New Zealand. AgResearch provided support in the form of a salary for author CP, but did not have any additional role in the study design, data collection and analysis, decision to publish, or preparation of the manuscript. The specific roles of this author is articulated in the 'author

eradicate at least 138 insect species [7]. About 285 attempts (24%) have targeted 27 Lepidoptera species, which have all been moths rather than butterflies [7].

In May 2010, the Palaearctic large white butterfly, *Pieris brassicae* L. (Lepidoptera: Pieridae), was detected for the first time in New Zealand in Nelson (Fig 1) [8]. It had previously been accidentally introduced to South Africa [9] and Chile [10], and may have reached Nelson via its seaport as pupae on imported shipping containers, which is a known pathway for *P. brassicae* [11,12]. The potential for *P. brassicae* to cause harm in New Zealand had been recognised since at least 2001 when it was listed as an Unwanted Organism under the New Zealand Biosecurity Act 1993. It was also predicted to be relatively likely to invade New Zealand [13].

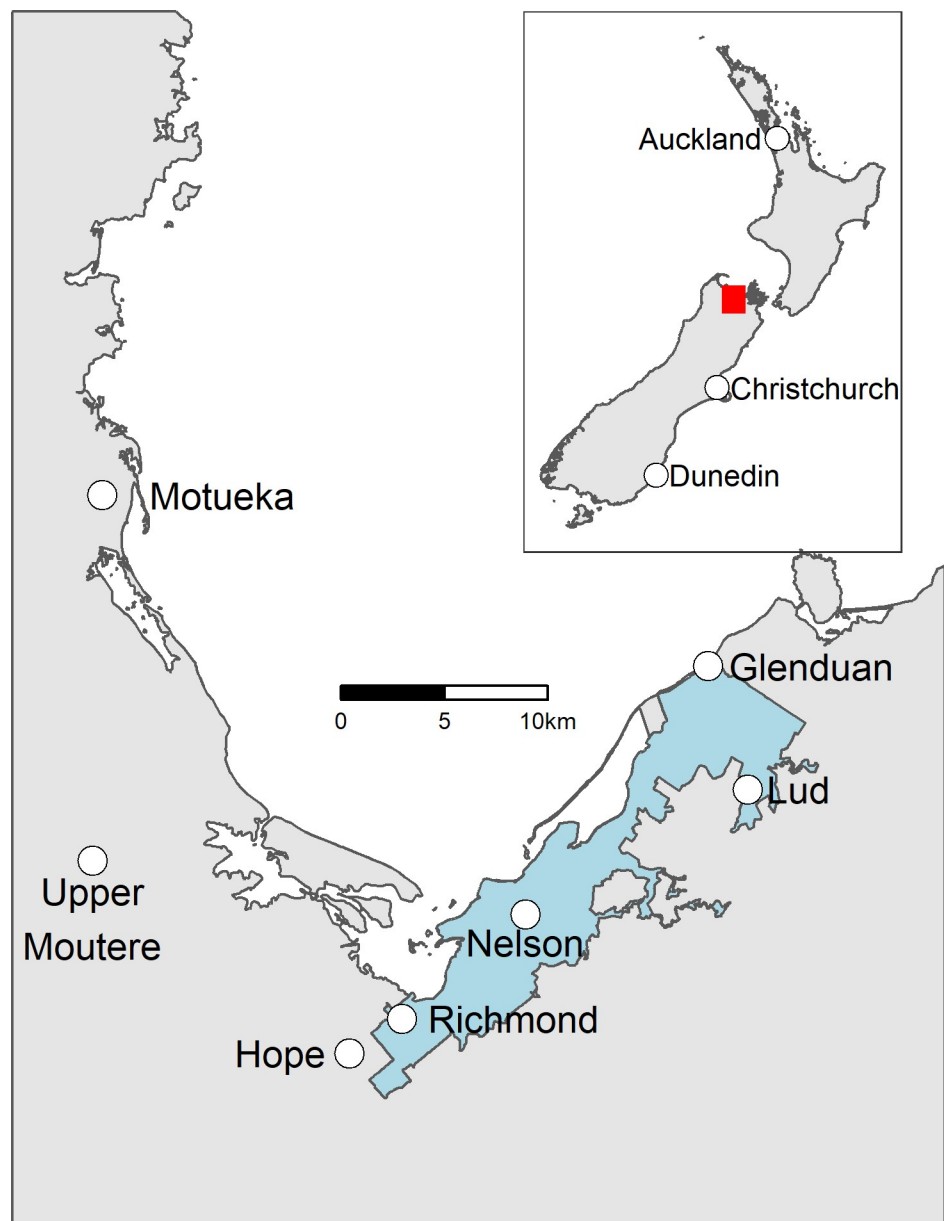

**Fig 1. Map of Nelson and its environs with the *Pieris brassicae* eradication operational area shaded in blue.** The red rectangle in the inset map indicates the position of the main map relative to the rest of New Zealand.

contributions' section. Plant and Food Research provided support in the form of a salary for author GW, but did not have any additional role in the study design, data collection and analysis, decision to publish, or preparation of the manuscript. The specific roles of this author is articulated in the 'author contributions' section. The New Zealand Ministry for Primary Industries (www.mpi.govt.nz) provided financial support for some of the research costs of CP, GW and RT, and the New Zealand TR Ellet Agricultural Trust contributed support for some of the research costs of CP. MPI and TR Ellet Agricultural Trust did not have any additional role in the study design, data collection and analysis, decision to publish, or preparation of the manuscript.

**Competing interests:** The authors have declared that no competing interests exist. RT is a the Managing Director of a commercial company EntEcol Ltd (www.entecol.co.nz) which provides technical entomological services to New Zealand clients. In the eradication program, EntEcol Ltd was contracted by DOC for RT to provide services including contributing to the TAG, preparing documents, identifying specimens, helping to develop the visual lure, and evaluating P. brassicae parasitism rates. EntEcol Ltd provided support in the form of a salary for author RT, but did not have any additional role in the study design, data collection and analysis, decision to publish, or preparation of the manuscript.Author RT's commercial affiliation to EntEcol Ltd does not alter our adherence to PLOS ONE policies on sharing data and materials.

In the northern hemisphere, *P. brassicae* larvae have been observed in the field feeding on at least 91 plant species from 12 plant families [14]. Sixty (66%) of these plants belong to the Brassicaceae and include cultivated and wild species [14]. A *P. brassicae* female lays about 500 eggs on host plants in batches of 50–150 eggs [15]. Larvae feed gregariously and may defoliate several plants during their development. Fifth instar larvae crawl away from their host plants to pupate, typically on vertical surfaces in sheltered locations [14].

The Ministry for Primary Industries (MPI) leads New Zealand's biosecurity system and is accountable under the Biosecurity Act 1993 to protect New Zealand's environment, economy, health and socio-cultural values from harmful organisms [16]. MPI responded to *P. brassicae* by formally identifying it in consultation with an expert lepidopterist (J. Dugdale, Manaaki Whenua Landcare Research, Nelson) [8], alerting the public, establishing a monitoring program to slow its spread and evaluating an eradication attempt. *Pieris brassicae* adults migrate long distances in Europe [17], which suggested it could spread quickly in New Zealand, and this impression was reinforced by *P. rapae* which took just 5–8 years to spread throughout New Zealand [18]. Surprisingly, however, *P. brassicae* still appeared to be restricted to Nelson 2 years after it was first recorded there [19]. Nevertheless, MPI terminated its response in November 2012 because it considered an eradication attempt would probably fail and the expected benefit to cost ratio was too small [16].

New Zealand's Department of Conservation (DOC) is responsible for protecting native biodiversity under the Conservation Act 1987 [16] and was concerned that *P. brassicae* could harm New Zealand native Brassicaceae, which comprise 3% of New Zealand's indigenous flora (S. Courtney, pers. comm., 2020). New Zealand has 83 native Brassicaceae species (81 endemic) in five genera, of which three genera—*Cardamine*, *Lepidium* and *Rorippa*—also contain northern hemisphere species that are fed upon in the wild by *P. brassicae* [14]. The remaining two New Zealand genera—*Notothlaspi* and *Pachycladon*—do not occur in the northern hemisphere [20], and their potential suitability as hosts for *P. brassicae* is less clear. Sixty six (80%) of New Zealand's 83 native Brassicaceae have received threat classifications under a New Zealand system that was adapted from the International Union for Conservation of Nature Red List [21]. Of the three genera that contain species fed upon by *P. brassicae* in the northern hemisphere [14], New Zealand has: 46 native *Cardamine* spp. of which 38 (83%) are threatened and 11 (38%) are nationally critical (the highest threat level); 21 native *Lepidium* spp. of which 20 (95%) are threatened and 12 (57%) are nationally critical; and three native *Rorippa* spp. of which one (33%) is classified as nationally vulnerable [22].

The close taxonomic relationships of these New Zealand native Brassicaceae to some northern hemisphere *P. brassicae* host plants indicated they could become novel hosts for *P. brassicae* should the butterfly spread more widely in New Zealand. Moreover, this concern was reinforced by knowledge that another closely related invasive butterfly, *P. rapae*, already damages wild populations of at least one of New Zealand's threatened *Lepidium* spp. [23]. The prospect of protecting New Zealand native Brassicaceae from herbivory by established populations of *P. brassicae* for the foreseeable future was infeasible because *P. brassicae*'s potential distribution was expected to extend throughout New Zealand [24] and many populations of New Zealand native Brassicaceae are tiny, spatially isolated and difficult for humans to access. *Pieris brassicae* was clearly also a threat to cultivated brassicas in New Zealand [14]. Thus, in November 2012 DOC began the first-ever attempt globally to eradicate a butterfly. The program's initial operational definition of eradication was: "Despite active searching, *P. brassicae* has not been detected for two consecutive years, or for a period statistically defined as providing high confidence that it has been eradicated" [25].

The operational details of many previous eradication programs reside in relatively inaccessible grey literature, which limits opportunities for learning [26–28]. This paper aims to inform future eradication programs by summarising the methods used and results obtained.

## Methods

All work described in this manuscript that involved human subjects was conducted with strict adherence to legislation described in the New Zealand Biosecurity Act 1993 (http://www.legislation.govt.nz/act/public/1993/0095/latest/DLM314623.html). The data were collected by staff from DOC and MPI who were authorised to do so under the New Zealand Biosecurity Act 1993. *Pieris brassicae* is legislated as an unwanted organism under this Act, which means authorised persons have a wide range of statutory powers to enable them to control it; including accessing, inspecting and applying treatments on privately owned properties.

We define a 'detection' as the discovery of one or more *P. brassicae* at one location at one time. Thus, detections refer to the number of inspections that revealed *P. brassicae* rather than to the number of *P. brassicae* individuals found.

### Management and review

A strategy was prepared before the eradication attempt commenced that documented the program's goal, objectives, actions, timeframes, stopping rules, and staff roles and responsibilities [29]. The program implemented a cycle of 'plan, implement, monitor, report and review', and emphasised team work, effective communication, and openness to suggestions for improvement (Table 1, S1 Text). A Technical Advisory Group (TAG) of six people with expertise in eradication and invertebrate ecology was assembled and led by DOC (author K. Brown), and produced plans, provided advice, conducted research, lobbied for financial support, and reported results (Table 1, S1 Text). The program was reviewed in August 2013 by DOC and in December 2013 by MPI. DOC's review sought to confirm the program was being well managed and identify opportunities for improvement [30]. MPI's review had similar goals plus it evaluated the program's likelihood of success [31,32] (S1 Text).

The TAG developed nine criteria to help evaluate and guide the eradication attempt [33], which it regularly used to steer discussion, qualitatively assess program feasibility and identify needed improvements. Though not designed to quantitatively estimate probabilities of eradication success [33], each year from 2013 to 2015 five TAG members and another expert were asked to use the criteria to independently evaluate the program and informally derive their own probability estimate: The range and mean of these estimates were then reported to managers. Progress was publicly reported via a series of annual reports [25,34–36].

### Operational area

An area of ca. 14600 ha was intensively managed during the eradication attempt and is termed the 'operational area'. It included Nelson City (41.29˚S, 173.28˚E), the adjoining urban area of Richmond, and farmland (Fig 1). It was populated by ca. 47000 people living in ca. 32000 households, and the main *P. brassicae* host plants present were brassica vegetables in home gardens, and nasturtium (*Tropaleum majus*) in gardens and wasteland. Some naturalised brassicas were also present [32]. Commercial brassica crops mainly occurred outside the operational area.

Nelson has a temperate oceanic climate with a summer average maximum temperature of 22˚C and a summer minimum of 12˚C. Winter average maximum and minimum temperatures are 14˚C and 4˚C. Average annual rainfall is 1043 mm, and average annual sunshine is

**Table 1. Summary of the critical components of the *Pieris brassicae* eradication program.**

| Strategy | Delimit population |
|---|---|
| | Contain |
| | Eliminate |
| | Monitor to confirm eradication |
| Management | Evaluate reasons to eradicate |
| | Assess feasibility |
| | Establish technical advisory group |
| | Plan |
| | Define stopping rules |
| | Define roles and responsibilities |
| | Train and motivate staff |
| | Encourage team work |
| | Structure decision making |
| | Engage and use scientific support |
| | Collect and analyse data |
| | Monitor results |
| | Communicate and report (internally and externally) |
| | Review |
| | Adapt |
| | Manage budget |
| Tactics | Engage with stakeholders |
| | Foster community support |
| | Identify locations of potential habitat/host plants |
| | Prioritise search locations |
| | Align search timing and frequency with pest phenology |
| | Systematically search and treat habitat |
| | Capture butterfly adults |
| | Conserve and augment natural enemies |
| | Monitor potential emigration |

2449 hours. Mountains border Nelson's eastern perimeter from the south to the northeast, ocean lies to the northwest, and to the southwest is an intensively farmed plain.

To facilitate management, the operational area was divided into 46 management blocks (S1 Fig) with areas ranging from 27–1944 ha (S1 Data). Within blocks, the units searched were mostly residential properties, though some commercial properties and public green spaces were also searched. Properties per block ranged from just 12 in a block that was predominantly farm land to ca. 2400 (S1 Data).

## Active surveillance

We define active surveillance as planned systematic searching for *P. brassicae* by DOC staff.

**Field staff.** All field staff underwent police vetting and employment checks prior to appointment and received Authorised Persons training to give them legal access to private properties without landowner permission under the New Zealand Biosecurity Act 1993. Training (Table 1) included communicating with property owners, managing aggressive dogs, first aid, identifying *P. brassicae* and its host plants [37], search methods, handling and applying pesticides, and data recording.

The eradication attempt began in November 2012 with only three field staff. As the scale of the eradication challenge became clearer, this number was increased to 24 by April 2013 and to 35 by November 2014. Field staff were divided into eight teams, each comprising 2–8 people. Six teams searched for *P. brassicae*, one specialised in controlling larger areas of host plants, and one responded to residents' reports of sightings and reinspected previously treated properties (Table 1). Teams were issued with VHF and UHF radios, and team leaders carried mobile phones. Each day, teams were assigned to search particular properties specified via analysis of previous surveillance results (see below).

**Prioritising locations to search.**   The program aimed first to eliminate *P. brassicae*, then to continue surveillance to confirm eradication (Table 1). During the elimination phase, the program prioritised the destruction of small peripheral *P. brassicae* populations to minimise spread beyond the operational area, while simultaneously treating the larger central population to reduce population growth and emigration pressure [38]. All properties in the operational area that had potential to contain host plants were repeatedly searched. Properties that were not regularly inspected included some in commercial and industrial areas with minimal vegetation.

In winter and summer when *P. brassicae* was relatively difficult to detect (see below), inspections sought to identify all properties in the operational area with host plants so these could be precisely targeted in spring and autumn when *P. brassicae* was easier to detect. The operational area was searched block by block, often with two or more blocks being searched simultaneously by separate teams. To attempt to maximise *P. brassicae* mortality, blocks were prioritised for searching (Table 1) based on their mean *P. brassicae* detection rates during the previous spring and autumn, plus factors such as logistics and season [39]. During elimination, locations where *P. brassicae* and its host plants had seldom been recorded were searched relatively infrequently and mostly in summer or winter.

The program's transition from elimination to monitoring (Table 1) demanded confidence that *P. brassicae* was absent from the entire operational area, including locations infrequently searched during the elimination phase. Again, the emphasis of spring and autumn searching for *P. brassicae* was on properties identified to have host plants during the previous winter or summer. Allocating search effort across all 46 blocks (S1 Fig) to maximise confidence *P. brassicae* had been eradicated was informed by a model that estimated relative probabilities of *P. brassicae* being present in each block (Kean and Phillips, in preparation).

**Search timing and frequency.**   The phenology of *P. brassicae* was modelled [40] using published data for its developmental responses to temperature [41] and day length [42]. The model was validated against observations of *P. brassicae* in both the northern hemisphere and New Zealand, and helped to define the timing and frequency of searches (Table 1).

*Pieris brassicae* had 2–4 generations per year in Nelson. Most *P. brassicae* overwintered as pupae, from which adults emerged in spring to lay eggs. In summer, approximately half of the population aestivated as pupae, with second generation adults emerging in autumn, which coincided with the emergence of third and fourth generation adults emerging from non-aestivating pupae [40].

*Pieris brassicae* pupae were difficult to find [43] and prevailed in summer and winter. During these seasons all blocks were surveilled for host plants to enable the highest risk properties to be targeted the following autumn or spring when other more detectable life stages predominated. Nevertheless, some searching for pupae was also conducted in winter (see below).

During spring and autumn, consecutive bouts of surveillance in the same location occurred at different intervals depending on if and when *P. brassicae* had been detected there [44]. In general, the program aimed to search properties in high priority blocks frequently enough to prevent any *P. brassicae* eggs laid after the previous search from becoming pupae before the

next search; ca. every 2–4 weeks. However, if *P. brassicae* was detected on a property, the property was searched again before any eggs overlooked in the previous search could reach the pupal stage; ca. every 1–2 weeks. Reinspections of infested properties usually continued until no *P. brassicae* had been detected in two consecutive inspections. These short interval reinspections enabled the efficacy of searches for *P. brassicae* eggs and larvae to be estimated [43].

**Search methods.** Properties were visited during the day and, if residents were present, permission to search was requested. If residents were absent, gardens were searched for *P. brassicae* and its host plants (Table 1), and notification of the search was left. When properties could not be searched (e.g., due to threatening dogs, locked gates or unhelpful residents), contact was made again by phone or letter and access arranged.

Eggs and larvae were sought by systematically inspecting all host plants. Any found were removed, then host plants were treated. Immature *P. brassicae* were either killed upon detection, or kept in captivity to monitor parasitism then killed.

Pupae were searched for throughout the year, but were explicitly targeted during winter on properties where mid–late stage larvae had been detected the previous autumn. Inanimate objects such as fences, garden sheds and house exteriors were searched using ladders and torches as necessary to inspect cracks and crevices. Adjacent properties were also searched if it was suspected that larvae had crawled off the property to pupate.

Adults were searched for in sunny locations with abundant nectar sources and captured with hand-held nets (Table 1). This was often difficult and time consuming due to *P. brassicae*'s rapid and evasive flight, but was considered worthwhile because: Capturing gravid females minimised the number of eggs they could otherwise have laid, potentially over many hectares; and capturing males when adult populations were low potentially inhibited mate finding and reduced female fecundity.

Research was conducted to develop attractants for *P. brassicae* adults, but did not produce practicably useful results [45,46]. However in 2014 a DOC staff member, W. Wragg, developed an ultra-violet (UV) reflective lure that was attractive to *P. brassicae* adults. Its efficacy was optimised by measuring the UV reflectivity of various materials [47] to identify one with similar reflectivity to *P. brassicae* wings [48,49]. A cloth with suitable UV reflectivity was glued to ornamental butterflies' wings, which moved by solar power, and the models were used to attract *P. brassicae* adults towards staff with nets.

## Passive surveillance

Publicity aimed to engender support for the eradication program and promote reports of *P. brassicae* (Table 1), and occurred at times when *P. brassicae* adults, eggs and larvae were about to appear. Communication methods included: DOC's website; a Facebook page; newspapers; magazines; billboards; leaflets and letters dropped in letter boxes; information displays and fridge magnet giveaways at events; face to face discussions with vegetable sellers and other groups; public talks; school visits; thank you cards to helpful property owners; newsletters regularly sent to stakeholders; advertisements at a local cinema; and advertisements, interviews and articles on local and national radio stations. Information given included descriptions of risks associated with: Accidentally moving *P. brassicae* pupae out of Nelson on vehicles such as campers and caravans, which are often stored near gardens; accidentally moving *P. brassicae* larvae out of Nelson on home-grown brassica seedlings, vegetables and vegetable waste; and use of brassicas as winter cover crops. Automobile mechanics were asked to be vigilant for *P. brassicae* pupae when conducting safety checks of vehicles, trailers, and caravans. Interpreters were employed to talk to recent New Zealand immigrants in their first language. The public were asked to report sightings of *P. brassicae* via a continuously monitored toll-free number

operated by MPI. Reports were immediately conveyed to DOC, which responded within 48 hours, usually visiting the properties for verification.

**Bounty hunt.** A NZ$10 bounty was offered for each dead *P. brassicae* adult given to DOC during a 2 week school holiday in spring 2013. The bounty was only offered for this one period to minimise motivation to culture *P. brassicae* for profit.

## Population delimitation

Monitoring for *P. brassicae* outside the operational area (Table 1) occurred via active surveillance, passive surveillance, monitoring of native brassica populations by DOC, and searching commercial brassica crops by staff from a nearby crop research institute, who searched for *P. brassicae* when conducting routine scouting for other pests in brassica crops.

## Treatments

**Insecticides.** A program review (Table 1) recommended that all *P. brassicae* host plants at a site should be sprayed with insecticide whenever eggs or larvae were found because search efficacy was likely < 100% [30]. Consequently, the BioGro-certified organic insecticide Entrust® SC Naturalyte® (active ingredient spinosad) was chosen because it was the most socially acceptable option and would have minimal impacts on *P. brassicae*'s insect natural enemies (Table 1; see below). The horticultural mineral oil D-C-Tron® was added to improve spray coverage and increase egg mortality. Spraying was usually conducted after gaining consent from property occupants, but occasionally occurred without consent when the occupants could not be contacted and late-stage larvae were found. If occupants resisted this treatment then one of the following alternatives were used: Either removing or regularly inspecting host plants, or applying a microbial insecticide, Dipel DF®, which contains toxins from the bacterium *Bacillus thuringiensis* (Bt) subspecies *kurstaki*.

Insecticides were applied following label directions by staff certified under the New Zealand Standard for Management of Agrichemicals (NZS 8409:2004) using either Solo® 15 L professional backpack sprayers, or Solo® 5 L and 7.5 L professional manual sprayers. Sprayers were fitted with brass adjustable nozzles (C-Dax Ltd) and ball valve filters. They were not calibrated because insecticide was spot-applied to host plants to the point of run off. Staff wore appropriate personal protective equipment including respirators with replaceable filters. Public notifications of spraying were not posted because most applications occurred on private land where owners had given consent and been notified, and the few applications made on public land were in locations that were difficult to access.

**Host plant control.** Host plant patches were prioritised for control based on their size and proximity to *P. brassicae* detections, and treated sites were reinspected to verify treatment efficacy. Staff with abseiling experience accessed host plants on steep terrain. Nasturtium growing in unpopulated areas was treated with a mixture of glyphosate, a desiccant (carfentrazone-ethyl), a surfactant, plus an insecticide (bifenthrin) in case any *P. brassicae* were present. Herbicides were applied as previously described for insecticides. When applying herbicides on steep slopes, including when abseiling, staff used the lighter 7.5 L sprayers carried in hiking backpacks to reduce weight.

**Biological control.** During the 1930s, two parasitic wasp species were introduced to New Zealand for biological control of *P. rapae*: *Cotesia glomerata* L. (Hymenoptera: Braconidae), which parasitises larvae, and *Pteromalus puparum* L. (Hymenoptera: Pteromalidae), which parasitises late-stage larvae and pupae [50]. Both species also parasitise *P. brassicae* [50] and were present in Nelson before *P. brassicae* was detected there.

Parasitism of *P. brassicae* by *C. glomerata* within the operational area was evaluated from October 2013 until June 2014 during active surveillance. *Pieris brassicae* larvae were subsampled (ca. 10 larvae per brood) and individuals were placed in separate pottles with brassica leaf for food then reared to fate (adulthood, death or parasitoid emergence) [51]. This work was conducted at a Nelson laboratory to avoid moving insects beyond the operational area.

To attempt to augment parasitism in the operational area (Table 1), *C. glomerata* cocoons were collected from *P. rapae* infestations in several New Zealand locations [51,52] and from *P. brassicae* infestations in Nelson. Cocoons were maintained until adult emergence, and adults were provided with 10% sugar solution via a vial with a cotton wick and allowed to mate. During autumn 2014 and autumn 2015, *C. glomerata* adults were released in locations where there had been either: Recent repeated *P. brassicae* detections; recent detections in areas that were difficult to search; or few recent searches. No attempt was made to evaluate if the releases increased parasitism rates.

In autumn 2015, laboratory cultured *Pt. puparum* were released as larvae developing within *P. rapae* pupae at locations where there was a high risk of *P. brassicae* late-stage larvae and pupae being present [53]. To measure if the releases increased parasitism rates, unparasitized sentinel *P. rapae* pupae were situated in cages accessible to *Pt. puparum* adults either within 2–3 m of the release locations, or > 200 m from them, then monitored for parasitism [53].

## Data collection and management

Data management (Table 1) was continuously refined and ultimately rested on a Geospatial Information System (GIS) built on an Environmental Services Research Institute ArcGIS Server. Web GIS (Geocortex Essentials) Version 4.4.2 was used to enter property inspection data. ArcGIS Version 10.3.1 was used to analyse spatial data and produce interactive maps, with dynamic queries indicating the highest priority properties to surveil. It was also used to help update the underlying Nelson cadastre to ensure that teams visited the correct addresses.

Field teams took a map of locations to be searched, conducted the inspections, and manually recorded details of any *P. brassicae*, host plants and access issues (S2 Fig). This information was transferred to the GIS typically within 48 hours and used to produce updated maps for subsequent surveillance (Table 1). A data analyst refined processes for data entry, capture, storage and analysis, and developed models that provided staff with access to reports on factors such as blocked access, safety (e.g. aggressive dogs), surveillance results, host plant control, and properties to be searched.

## Data presentation

Data were manipulated and Figs 1–3 created using the statistical programming language R version 3.6.0 [54] and functions in the R packages 'tidyverse' [55], 'sf' [56] and 'ggsn' [57]. Figs 1 and 3 used data sourced from the Land Information New Zealand Data Service licensed for reuse under CC BY 4.0.

## Results

### Management and review

The September 2013 feasibility assessment [58] concluded that seven of the nine criteria [33] were being substantially met whereas two were only being marginally met: These were (i) *Irrespective of its density, the population can be forced to decline from one year to the next*, and (ii) *Immigration and emigration can be prevented.*

DOC's August 2013 review made recommendations, all subsequently implemented, to increase insecticide use on infested properties, prepare a formal communication plan, and increase public awareness and community involvement in the program [30]. MPI's December 2013 review concluded that the program was being appropriately managed, it was too early to evaluate feasibility, and the program was worth continuing, but was concerned about *P. brassicae* escaping from the operational area [59].

An October 2013 estimate of the program's probability of success had a mean of 56% (range 50–60%, n = 6). However, the estimates increased in November 2014 to 80% (range 70–92%, n = 6) and in July 2015 to 91% (range 81–98%, n = 6).

## Active surveillance

Repeated inspections of infested properties enabled the efficacy of searches for *P. brassicae* to be estimated [43]. Following a single inspection, the proportion of properties where eggs or larvae were detected during the subsequent inspection declined from 32–52% in April–May 2013 when most staff were inexperienced to 5–25% in September–October 2013 when staff were fully trained. After late 2013 when insecticide use on infested properties increased, the proportion of properties where some *P. brassicae* eggs or larvae remained after an inspection declined to 1–11%. Thus, an insecticide treatment plus just one follow up inspection were sufficient to ensure all eggs and larvae had been eliminated from ≥ 99% of infested properties [43]. However, the program generally maintained two follow up inspections to maximise treatment efficacy.

Early in the program, field staff suspected that infested properties occurred in clusters with radii ca. 50–250 m. Thus, when *P. brassicae* was detected on a property, an early practice was to also inspect adjacent properties within these radii [25]. However, a spatial analysis of surveillance data found no evidence for clustered detections, thus it was concluded that searching properties that surround an infested property was unlikely to increase detection rates above searching randomly chosen properties in the same block [60] and the practice was discontinued. Further evidence that individual *P. brassicae* females often oviposited in disparate locations 2–5 km apart was obtained by analysing genetic variation in the mitochondrial COI gene of all detected specimens [61]. Because the location and life stage of every detected specimen had been recorded, the spatial distributions of potential offspring of each captured female could be modelled by matching the mitochondrial genotypes of female and immature *P. brassicae* while assuming a range of values for female longevity (Phillips, Sawicka and Kean, unpublished).

The UV lures were first deployed in October 2014 when detection rates had already declined to low levels (Fig 2). *Pieris brassicae* adults approached the lures in a manner similar to *P. rapae* [48,62], but never alighted on them. From 10 October 2014 to 3 November 2014, it took 180 person-hours to capture three *P. brassicae* adults without a lure, whereas it took 44 person-hours to capture seven with a lure.

According to DOC's June 2015 version of the Nelson cadastre, there were 32079 properties (total area 14614 ha) within the operational area and a further 9386 properties beyond it (total area 65054 ha; S1 Data). Field staff conducted 261962 inspections within the operational area and a further 2037 beyond it, giving a total of 263999 inspections. Of these, 111159 (42%) detected *P. brassicae* host plants, and 2884 (1%) detected *P. brassicae*; only three detections occurred beyond the operational area (S1 Data). Of the 32079 properties within the operational area, ca. 28730 (90%) had potential to contain host plants and were inspected an average of eight times during the program (S1 Data). *Pieris brassicae* host plants were detected at least once during the program on ca.17165 (60%) of the 28730 inspected properties (S1 Data). The

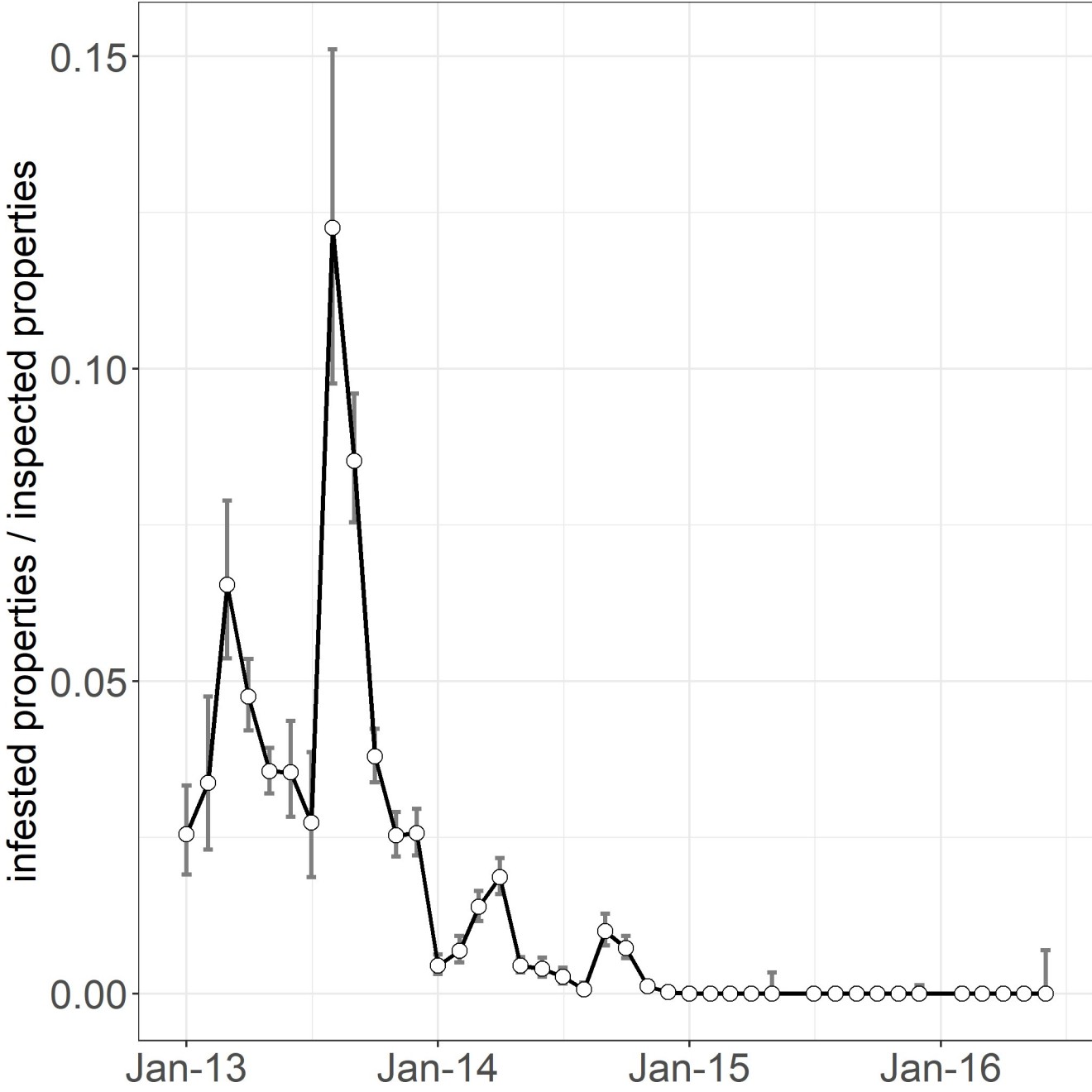

**Fig 2. Monthly *Pieris brassicae* detection rates from February 2013 to June 2016.** Error bars show 95% binomial confidence intervals.

most abundant host plant in Nelson was nasturtium and ca. 35% of detections occurred on this plant [25]. A similar proportion of detections occurred on broccoli, even though it was recorded less frequently in Nelson, which suggested it was a preferred host [25].

### Passive surveillance

A bounty for *P. brassicae* was offered for 2 weeks in spring 2013. In all, 319 individuals or groups handed in 3268 adults comprising 133 *P. brassicae* (4%) and 3135 *P. rapae* (96%) [32].

The *P. rapae* were from locations up to 130 km from Nelson, whereas *P. brassicae* only came from within the operational area.

The public submitted 1936 reports (additional to the bounty) of which 586 (30%) proved to be *P. brassicae* [34]. Most reports (76%) were made via the toll-free number, and the remainder were largely reported by phone directly to DOC's office in Nelson [34].

## Temporal changes in spatial distribution

*Pieris brassicae* was first detected in May 2010 and by October 2010 it had been found at eight properties in urban Nelson up to 12 km apart [63]. Over the next 2 years, passive surveillance reports suggested its distribution had not dramatically changed [19] (Fig 3, 'Before 1 Dec 2012').

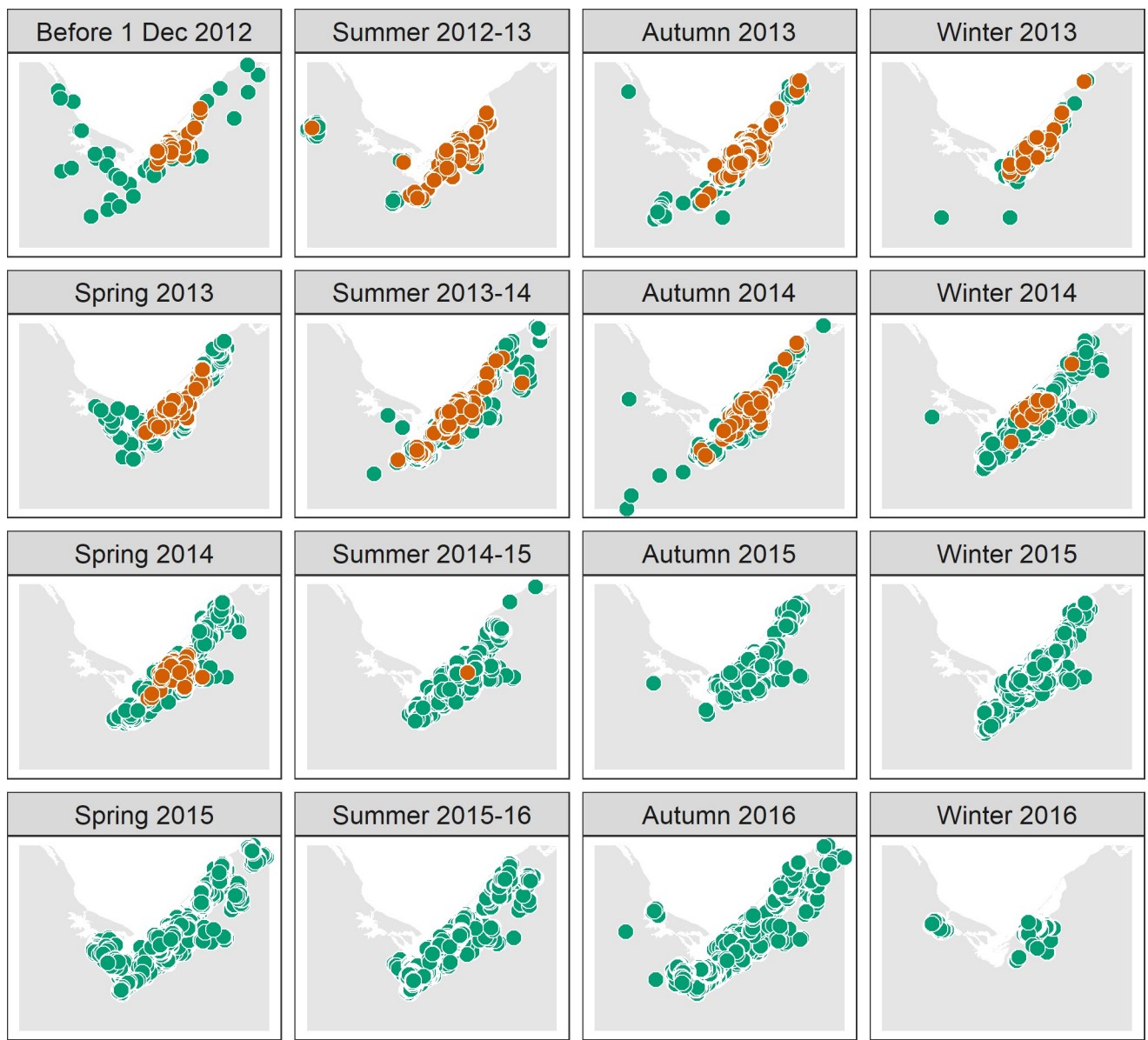

**Fig 3. Spatial distribution of *Pieris brassicae* from May 2010 to June 2016.** Green markers show search locations where *P. brassicae* was not detected and red markers (always plotted on top of green markers) show locations where it was detected.

When the eradication program began in summer 2012, there were several detections on the fringe of, or outside, the operational area. In summer 2012–13 (Fig 2), one (parasitised) *P. brassicae* larva was found ca. 25 km west of Port Nelson at Upper Moutere (Fig 1). This required intensive work to gain confidence additional *P. brassicae* had not escaped from the operational area, including increased publicity between Upper Moutere, Motueka and Nelson (Fig 1). The larva was likely taken to Upper Moutere from Nelson on an infested cabbage. Between autumn 2013 and autumn 2014 (Fig 3), several *P. brassicae* were detected ca. 11 km north of Port Nelson at Glenduan (Fig 1), which were managed via 895 inspections of 314 properties in that area. In summer 2013–14 (Fig 3), one adult was detected ca.15 km southwest of Port Nelson at Hope and another was detected ca. 10 km northeast of Port Nelson at Lud Valley (Fig 1). Intensive searching in the vicinities of these detections revealed no further *P. brassicae*.

Despite such dispersal events, from autumn 2014 *P. brassicae* became increasingly confined to central Nelson (Fig 3), and it became apparent during 2016 that the last detection had occurred near central Nelson in summer 2014–15 (Fig 3). Thereafter, active surveillance persisted until winter 2016 when confidence that *P. brassicae* had been eliminated was sufficient to terminate the program (Fig 3).

## Temporal changes in detection rates

Eggs, larvae and adults of *P. brassicae* were more detectable than pupae, thus there were peaks in detection rates during spring and autumn when they were more prevalent than pupae (Fig 2). Monthly rates peaked in September (spring) 2013 when *P. brassicae* (including all life stages) was detected on 9% of 2931 inspected properties. By this time, staff had been fully trained, *P. brassicae* was relatively abundant, and most of the population was exposed to control (i.e., few pupae). Thereafter, rates generally declined, though they showed regular smaller peaks each autumn and spring (when there were relatively few pupae) until the end of 2014. They declined to zero in January 2015 and remained there until 4 June 2016 when surveillance ended (Fig 2).

## Treatments

**Insecticides.** Following a detection, ca. 30% of property owners asked for an alternative treatment to Entrust® SC Naturalyte®: About 20% chose host plant removal, 5% chose regular host plant checks, and the remainder chose Bt [35].

**Host plant control.** To minimise potential concerns to residents, host plants were always treated from the ground rather than aerially, and were controlled on a mean of 2620 ± 489 (± SD) properties per year, with some properties treated up to three times annually to manage regrowth. Specialist abseiling skills and/or commercial herbicide sprayers were needed to apply treatments on ca. 15 properties per year. Nasturtium and other naturalised brassicas such as wallflower (*Erysimum* spp.) most often required specialist attention, with patches of up to 500 m$^2$ present in some steep locations.

**Biological control.** Monitoring of *C. glomerata* parasitism of *P. brassicae* during October 2013–June 2014 revealed that 65% of *P. brassicae* broods (n = 130) contained *C. glomerata*, and a mean of 35% of larvae (n = 999) per brood were parasitised [51]. To augment parasitism, ca. 10000 *C. glomerata* adults were released in the operational area during autumn 2014 and a further ca. 6600 were released in autumn 2015, though it is unknown if this increased parasitism rates [35].

During autumn 2015, over 14000 *Pt. puparum* adults were released at 17 Nelson properties [53]. Parasitism of sentinel *P. rapae* was rare—as were detections of *P. brassicae* pupae—and no effect of the releases on parasitism rates by *Pt. puparum* was measured [53].

## Data collection and management

Early data entry issues included a GIS interface that allowed users to inadvertently enter incorrect/invalid inspection dates and misspelled addresses, and provided users with inadequate confirmation that new records had been successfully entered and saved, which often provoked duplicate entries. These issues were compounded by the Nelson cadastre initially being incomplete and out of date, which sometimes created confusion for field staff about the spatial locations of addresses and resulted in inspection records being assigned to incorrect addresses. These problems created a dataset that was time-consuming to correct before it could be reliably used for analysis. In November 2014, a data manager with GIS expertise was assigned full time to the eradication program, and remaining issues with the cadastre and GIS interface were resolved by early 2015.

## Program end

The attempt to eradicate *P. brassicae* ceased on 4 June 2016 [16]. At this time, neither the absence of any *P. brassicae* detections during 18 months of active searching nor any statistical modeling had strictly met the programs' initial operational definition of eradication [25]. However, during 2016 DOC was having increasing difficulty funding the program and MPI, which had legal responsibility for determining if New Zealand could be declared free of *P. brassicae*, became convinced by the program's surveillance data that the butterfly had been eradicated: That any remaining *P. brassicae* would have completed more than three generations during the 18 months between the last detection and program cessation was particularly compelling. *Pieris brassicae* was officially declared eradicated from New Zealand on 22 November 2016 [64,65], 6.5 years after it was first detected and 4 years after the eradication attempt commenced, thus becoming New Zealand's 69th successful arthropod eradication [7].

## Discussion

We have described the methods and results of a successful *P. brassicae* eradication program in the hope they will be useful to future attempts to eradicate other pests. At the heart of the program were simple, manual treatments applied during repeated searches of the operational area for *P. brassicae* and its host plants. These surveys helped to both limit the pest population and inform future priorities. The searching was complemented by public reports of sightings, which the program vigorously promoted. Here, we discuss elements of the *P. brassicae* program that assisted this straightforward approach to succeed, and some that inhibited it. We also describe some attributes of the program that should be replicable in many future eradication attempts.

Sometimes when nonnative organisms are discovered in new regions, little technical information is available to assist effective responses [66]. However, numerous studies of *P. brassicae* in its native range were available to support aspects of the eradication attempt including species diagnosis, identifying effective chemical treatments, defining the butterfly's host range and natural enemies, and developing a phenology model and lure. The comprehensive literature will also have contributed to the 2001 declaration of *P. brassicae* as an Unwanted Organism in New Zealand under the Biosecurity Act 1993: This was significant because it gave authorised staff the legal right to search and treat private properties for *P. brassicae*, and some DOC staff had this authorisation before the program began, which expedited training to authorise

additional staff. Unfortunately, it was not used to develop preparedness plans prior to the establishment of *P. brassicae* in New Zealand, which might have further increased the probability of eradication success [66].

*Pieris brassicae* eggs, larvae and adults are relatively conspicuous, and its eggs and larvae were distinctive among New Zealand insects. Moreover, *P. brassicae* eggs and larvae occurred on low growing, readily accessible host plants and larval feeding damage became more conspicuous as defoliation proceeded. These attributes increased the practicality and efficacy of manual searches, and would also have helped to foster public reports of sightings [67].

The program engendered strong public support and received valuable reports of sightings that accounted for ca. 20% of all *P. brassicae* detections. This was promoted by comprehensive publicity, rapid responses to reports, respectful and communicative staff, and the availability of an effective organic insecticide which was more acceptable to many residents than synthetic chemical alternatives. The bounty particularly excited public interest, plus it eliminated some *P. brassicae* and provided independent evidence that the population had been correctly delimited.

Numerous *P. brassicae* natural enemies were present in Nelson and probably facilitated population suppression. These included: The insect parasitoids *C. glomeratus* and *Pt. puparum* [50]; and insect predators such as *Vespula vulgaris*, *V. germanica* [68], *Polistes chinenis antennalis* [69], various species of ants [70], spiders, harvestmen and predatory beetles [71] and birds [72]. Moreover, several pathogens infect *P. rapae* in New Zealand [73,74] and some *P. brassicae* larvae and pre-pupae collected to evaluate parasitism rates exhibited symptoms consistent with granulosis virus infection (G. Walker, personal observation). The butterfly's potential population growth rate in Nelson was also limited by a proportion of the population entering aestivation, which reduced that part of the population's annual number of generations [40,75].

Throughout the program, doubt persisted that the feasibility criterion *Immigration and emigration can be prevented* [33] could be met. The possibility that people would accidentally carry *P. brassicae* immatures beyond the operational area (e.g., on infested host material) and the ability of *P. brassicae* adults to fly long distances [17] meant there was constant potential for the pest to escape the operational area and establish elsewhere. This risk was partly mitigated by both comprehensive publicity and assiduous treatment of pest populations on the periphery of the operational area. Nelson's topography probably also helped to reduce emigration rates because ocean lies to its northwest, the mountains to its east contained few host plants, and arguably the sole benign pathway for natural dispersal was across the agricultural plains to its south. Moreover, the abundant and diverse *P. brassicae* natural enemies in New Zealand might have reduced the chance that emigrants could found new populations due to biotic resistance [76–78].

Although the eradication attempt was assisted by numerous factors, it still presented many ecological, technical and operational uncertainties [16] and, like most other eradication programs, was complex [79,80]. Quantifying benefits and assessing feasibility are important prerequisites to commencing an eradication program [16,79,81]. With *P. brassicae*, an inability to measure the conservation values at risk in dollar terms and uncertainty about feasibility delayed the program's commencement by 2.5 years [16] even as *P. brassicae* population growth was increasing the eradication challenge. Nevertheless, the delay between detection and program commencement was less than the threshold of about 4 years beyond which eradication success becomes much less likely, as identified from a meta-analysis of 173 eradication programs [66].

Unlike many other successful eradication attempts in New Zealand and elsewhere, powerful detection tools such as pheromone traps were unavailable for *P. brassicae*, and detection relied

on searching. A meta-analysis of 672 arthropod eradication attempts [82] found that programs without sophisticated detection methods had low success probabilities, though this effect became non-significant when programs directed against two species that can be trapped using pheromones, *Lymantria dispar* (n = 73 programs) and *Ceratitis capitata* (n = 56), were excluded from analysis. The lack of powerful attractants for butterflies, which unlike moths use vision rather than long range sex pheromones to find mates [83], may have contributed to the dearth of previous attempts to eradicate butterflies [7]. Nevertheless, New Zealand conservationists, particularly DOC, have had many successes eradicating other organisms such as mammalian pests for which there are few powerful detection tools [84–86].

The data management issues experienced predominantly during the first 2 years of the program reduced operational and analytical efficiency, but did not create serious doubt about achieving the feasibility criterion, "*Programme is effectively managed, and its status is reliably monitored and accurately recorded*" [33]. This was because it was always apparent that the data were being collected and corrected. However, the inefficiencies suffered would probably have been avoided by employing a qualified full-time data manager with access to a suitable GIS from the outset.

The program began just as DOC was being restructured, which disrupted internal communication, created uncertainty about roles and budgets, and distracted managers. This culminated in the program receiving inadequate funding during January–June 2015 and being forced to reduce field staff, whose numbers were approximately halved during February–March 2015, then cut to zero during May–June 2015. However, in July 2015 the program's budget was renewed, many of the program's former field staff returned, and the eradication attempt recovered from what was widely perceived as a dire threat to its success. It subsequently became apparent that the last detection of *P. brassicae* had already occurred on 16 December 2014 and, critically, the renewed funding enabled the species' absence from Nelson to be demonstrated.

Several elements of the *P. brassicae* eradication program that we regard as vital to its success (Table 1) should also be replicable in future eradication attempts. Effective program management is essential [87–89] including excellent planning, leadership, administration and data management, and emphases on fostering assiduous field work, team spirit, role flexibility, open communication and an 'eradication attitude' [87,90]. Maintaining close relationships with scientists, encouraging their involvement and valuing their recommendations was key, as was effective public engagement. Although the individual effects of the various treatments applied to *P. brassicae* are unknown due to confounding, we nevertheless suggest that attempting to deploy multiple tactics that together put every insect life stage at risk is worthwhile [91], including those such as capturing adults that at low pest densities may contribute to demographic Allee effects [78].

## Supporting information

**S1 Text. Additional management details.**
(DOCX)

**S1 Fig. Nelson management blocks.**
(PDF)

**S2 Fig. Data recording form.**
(DOCX)

**S1 Data.**
(XLSX)

## Acknowledgments

We thank the following DOC staff members for their patient and persistent efforts: Neil Clifton, who gave the go-ahead despite the uncertainty; Bruce Vander Lee and Mike Shephard, who were Project Managers; Simon Bayly and Julie Murphy, who were Operations Managers; James Reid, who built the GIS database; Jo Rees, who led planning; Senay Senait and Kath Henderson, who managed the data; Nicola Gourlay, Eva Pomeroy and Rosemary Vander Lee, who managed logistics/ administration; Jaine Cronin, Sally Leggett and Trish Grant who made major contributions to community engagement; Dan Chisnall and Derek Brown, who managed host plant control; Keith Briden, who helped to review the program; and over 60 people who worked in the field. We also acknowledge the many people from MPI, AgResearch, Better Border Biosecurity (B3), Plant & Food Research and Vegetables New Zealand who generously provided their time, advice, support and expertise, particularly John Kean, Ela Sawicka, Chikako Van Koten and Nicky Richards (AgResearch/ B3), and Bruce Philip, Erik Van Eyndhoven and Susanne Krejcek (MPI). We also thank John Dugdale (Landcare Research) and Henk Geertsema (University of Stellenbosch) for their valuable advice, and the experts who reviewed the program in December 2013: Mandy Barron (Manaaki-Whenua Landcare Research), Jacqueline Beggs (University of Auckland), Ecki Brockerhoff (Scion), Stephen Goldson (AgResearch), Mark Hoddle (Center for Invasive Species Research, UC-Riverside, USA), Margaret Stanley (University of Auckland) and Patrick Tobin (USDA Forest Service, USA). Oluwashola Olaniyan (Lincoln University) and four anonymous reviewers provided helpful suggestions that improved the manuscript.

## Author Contributions

**Conceptualization:** Craig B. Phillips, Kerry Brown, Chris Green, Richard Toft, Graham Walker, Keith Broome.

**Data curation:** Craig B. Phillips.

**Formal analysis:** Craig B. Phillips.

**Funding acquisition:** Craig B. Phillips, Kerry Brown, Richard Toft, Graham Walker, Keith Broome.

**Investigation:** Craig B. Phillips, Kerry Brown, Chris Green, Richard Toft, Graham Walker.

**Methodology:** Craig B. Phillips, Kerry Brown, Chris Green, Richard Toft, Graham Walker, Keith Broome.

**Project administration:** Kerry Brown, Chris Green, Keith Broome.

**Resources:** Kerry Brown.

**Supervision:** Kerry Brown, Chris Green, Keith Broome.

**Validation:** Craig B. Phillips.

**Visualization:** Craig B. Phillips.

**Writing – original draft:** Craig B. Phillips, Kerry Brown.

**Writing – review & editing:** Craig B. Phillips, Kerry Brown, Chris Green, Richard Toft, Graham Walker, Keith Broome.

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
