## [Decision Letter · Decision Letter 0]

24 Jan 2020

PONE-D-19-32860

Eradicating large white butterfly from New Zealand eliminates a threat to endemic Brassicaceae

PLOS ONE

Dear Dr. Phillips,

Thank you for submitting your manuscript to PLOS ONE. After careful consideration, we feel that it has merit but does not fully meet PLOS ONE’s publication criteria as it currently stands. Therefore, we invite you to submit a revised version of the manuscript that addresses the points raised during the review process.

This article has been evaluated by two reviewers. Both of them recognize the merit of this work and its value as guide for future protocols for the control of invasive species. Both reviewers also agree that this paper should have a place in the main literature. However, while one of the reviewers was very positive about publishing the work in this journal, having only some minor comments or suggestions, the other reviewer had strong concerns with that. This reviewer’s main concerns were that (1) the work presented is not a proper research article, since there is no hypotheses testing, and (2) some of the presented results have been published before. Due to such contrasting reviews of the manuscript, I have had some discussions about it with other Staff Editors of the journal. We think that although the manuscript is not the typical research paper, it might be in scope for the journal because it quantitatively measured and reports the progress of the eradiation actions and the approach might be reproducible and the results generalisable/applicable to different contexts.

Therefore I would like to give you the opportunity of making a major revision of your work in which you consider all the important concerns raised by this reviewer. Regarding the question about dual publication, and in accordance with PLOS ONE publication criteria, we do not consider for publication any manuscript that has been formally submitted or published in the peer-reviewed literature (https://journals.plos.org/plosone/s/criteria-for-publication#loc-2). If part of this work was previously reported in conference proceedings or governmental reports this would not preclude it from publication in PLOS ONE, but we would like you to clarify in the response letter which results were previously published and where, and how this new work adds to previous publications. Please, upload also as supplementary information for review any previous work that you have published with these data. Also, I see that your figures show how the butterfly was eradicated, but I would like to see the quantitative impact of each particular action in time supported by statistical analyses. In this way, the efficiency and the timing of each action could be more objectively evaluated. We could consider the paper for further evaluation as long as these criticisms/comments are considered in deep and you can provide detailed and convincing answers to the concerns raised.

We would appreciate receiving your revised manuscript by Mar 09 2020 11:59PM. To enhance the reproducibility of your results, we recommend that if applicable you deposit your laboratory protocols in protocols.io, where a protocol can be assigned its own identifier (DOI) such that it can be cited independently in the future. For instructions see: http://journals.plos.org/plosone/s/submission-guidelines#loc-laboratory-protocols

We look forward to receiving your revised manuscript.

Kind regards,

Amparo Lázaro, PhD

Academic Editor

PLOS ONE

Journal Requirements:

'The authors have declared that no competing interests exist.'

We note that one or more of the authors are employed by a commercial company: Entecol Ltd

3. We noted in your submission details that a portion of your manuscript may have been presented or published elsewhere:

'None of the results in this manuscript have been taken from other published or pending manuscripts. However, many of the results have been presented in NZ Department of Conservation annual reports and various other non-peer-reviewed unpublished reports (i.e. 'grey literature').'

Please clarify whether this publication was peer-reviewed and formally published.

If this work was previously peer-reviewed and published, in the cover letter please provide the reason that this work does not constitute dual publication and should be included in the current manuscript.

Reviewers' comments:

Reviewer's Responses to Questions

**Comments to the Author**

1. Is the manuscript technically sound, and do the data support the conclusions?

Reviewer #1: Yes

Reviewer #2: No

2. Has the statistical analysis been performed appropriately and rigorously? 

Reviewer #1: Yes

Reviewer #2: I Don't Know

3. Have the authors made all data underlying the findings in their manuscript fully available?

Reviewer #1: Yes

Reviewer #2: No

4. Is the manuscript presented in an intelligible fashion and written in standard English?

Reviewer #1: No

Reviewer #2: Yes

5. Review Comments to the Author

Reviewer #1: First, I'd like to congratulate the authors on a tremendously successful project, the world's first eradication of an invasive butterfly that presented a risk not only to agricultural crops, but also to endangered and endemic native NZ brassicas. Second, the paper is very well written, I enjoyed reading it, and I didn't feel compelled to do a lot of editing to improver readability/flow, and third, a rich trove of insight and experiences are provided here in addition to reporting of process and outcomes. These insights are extremely valuable.

My recommendation is to accept and publish with minimum revisions.

One area I would like significant improvement is on pesticide applications - these are critical in eradication projects, but the deployment and use of this tool has not been covered adequately. Please respond to all queries in the "comments" call out on the attached PDF of this article.

Additionally, there are a lot of "grey" references cited that appear to be almost irretrievable.

I've highlighted a lot of these in pink in the attached PDF - I may have missed some, but I think I've selected sufficient to give the idea of the types of materials that I am concerned about.

My feeling from reading the paper is that these "grey" documents have very useful information, hence their citation, and they need to be accessible for future use. Since they have been cited, the information is obviously not confidential. I think these materials need to be web accessible, either via supplemental materials associated with this paper, or via a "Giant White Butterfly Eradication Resource" website, possibly maintained by DOC(??!?!??)

Finally, perhaps in the discussion. I think it would be instructive to cover these three things: (1) the slow spread of the butterfly around Nelson - could this be due to a "lag" event - hence the critical 4 year window cited in the discussion? (2) The DNA work - can this be used to get a feel for founding population sizes and possible inbreeding that could've affected fitness (and subsequent rates of spread?), finally, Tobin/Liebhold/Suckling have made compelling arguments that you don't need to eradicate every last individual, you just need to get the population low enough so that destabilizing Allee effects can act and help push the population to extinction - do the authors have any thoughts on how Allee effects may have played a role in this program?

One last thing, biotic resistance is captured in the discussion, parasitoids/predators, no mention of pathogens (e.g., viruses/bacteria/fungi/microsporidia), any field observations to report on this guild of natural enemies? Any evidence for microsporidia in the invasive population that could have affected fitness??

Overall, this is a great piece of work, the outcomes are excellent, and the paper is really well put together.

Reviewer #2: The paper describes operational efforts to eradicate the large white butterfly, Pieris brassicae, from New Zealand. It is not a research paper (i.e., no hypotheses are tested). The paper may provide a model for others to follow in the event of a future pest incursion. I believe such topics have a place in the primary literature. Clarification of a number of issues would help to make the paper more useful:

1) The paper needs to more clearly acknowledge other published summaries of the same eradication program and explain the unique contribution that is being made herewith. For example, Brown et al. (2019; cited by the authors) provide a thorough overview of the eradication program and seem to summarize the same data. (It appears that small portions of the Brown paper are quoted verbatim or closely paraphrased without appropriate attribution; e.g. lines 73-75; 86-88.) Likewise, Figures 2 and 3 in the current paper seem remarkably similar/identical to figures in Phillips et al. (2016; cited by the authors).

2) The true risk that P. brassicae posed/poses to native brassicas of New Zealand is not clear. It is not clear whether a formal pest risk assessment was ever completed for the species. Brown et al. (2019) note that the species was “[a]n Unwanted Organism under the Biosecurity Act of 1993, and a known pest of cultivated brassicas …”. The logic in the current paper seems to be that P. brassicae feeds on many species within the Brassicaceae; therefore, all Brassicaceae are at some risk of herbivory. Of course, this position is not supported empirically (i.e., oligophagous herbivores are not able to recognize and develop on ALL members of a plant family, Menken et al. 2010. Evolution 64:1098). Brown et al. (2019) provide some justification for the risk-averse position. However, a formal analysis could be done to compare the phylogenetic distance of plants of concern from known hosts. Such an analysis would be useful to justify the title of the paper, if it demonstrated that New Zealand natives were close evolutionarily to known hosts.

3) The authors highlight the fact that this effort appears to be the first attempted eradication of a butterfly. No explanation is given as to why this is significant. Is eradication of a butterfly expected to be significantly more/less difficult than the eradication of moths? Why?

4) The methods begin with a significant discussion of the administrative structure of the eradication program. It is not clear which components of this structure should be considered essential. Many details seem better suited for supplementary details. More emphasis should be given to critical details, for example, a synopsis of the nine criteria and methods used to assess quantitatively the probability of eradication success. Phillips et al. (2019) review the criteria but do not describe how to use them to generate a probability of eradication success. Expert opinion appears to play a prominent role, but this is not specified.

5) The authors do not provide an operational definition for ‘eradication’ at the outset. In the United States, three insect generations with intensive trapping and no catches are required before eradication can be declared. It is not clear whether this standard was met in New Zealand. The non-zero error bar at the end of the program in the current Fig 2 suggests that at least one P. brassicae might have been detected.

6) The structure of the eradication program lacks clarity. I consulted the Additional Information 2, which provides a map of the 46 management blocks. How much potentially suitable habitat for P. brassicae occurs within each block, how many total properties have suitable habitat within each block, and how many properties within each block were sampled? These are all critical details. The authors do not describe how data were analyzed. As the inspections do not appear to represent a simple random sample, the simple proportion of inspected properties that were infested is not an appropriate response variable. From the description, it appears that monitoring followed a stratified sampling scheme and resulting data should be analyzed accordingly.

7) Given that insecticide treatments were highly-localized to individual host plants, it seems likely that some portion of the population would go untreated in each generation. Other eradication programs have relied on area-wide insecticide applications to achieve success. What proportion of the P. brassicae population would need to be treated in each generation to drive the population to zero in less than two years? Was such a level achievable? Could other factors (e.g., parasitism) have contributed to the success of the eradication?

8) In the end, it would be helpful if the authors could comment on which aspects of the eradication effort would be most replicable in other situations.

6. PLOS authors have the option to publish the peer review history of their article (what does this mean?). If published, this will include your full peer review and any attached files.

Reviewer #1: No

Reviewer #2: No

---

## [Author Response · Author response to Decision Letter 0]

26 Mar 2020

We have provided comprehensive responses to all of the editor and reviewer comments in our 'response_to_reviewers.doc' uploaded with the revision. The text from that document is cut and pasted below, though it is more difficult to interpret than the word document because the formatting has been lost.

Questions regarding prior publications and our responses

Regarding the question about dual publication, and in accordance with PLOS ONE publication criteria, we do not consider for publication any manuscript that has been formally submitted or published in the peer-reviewed literature (https://journals.plos.org/plosone/s/criteria-for-publication#loc-2). If part of this work was previously reported in conference proceedings or governmental reports this would not preclude it from publication in PLOS ONE, but we would like you to clarify in the response letter which results were previously published and where, and how this new work adds to previous publications. Please, upload also as supplementary information for review any previous work that you have published with these data.

We noted in your submission details that a portion of your manuscript may have been presented or published elsewhere:

'None of the results in this manuscript have been taken from other published or pending manuscripts. However, many of the results have been presented in NZ Department of Conservation annual reports and various other non-peer-reviewed unpublished reports (i.e. 'grey literature').'

Please clarify whether this publication was peer-reviewed and formally published.

If this work was previously peer-reviewed and published, in the cover letter please provide the reason that this work does not constitute dual publication and should be included in the current manuscript.

Authors:

Below is a list of all 7 previously published works related to the New Zealand (NZ) P. brassicae eradication program. As requested, they have all been uploaded as supplementary information. Items 1 – 4 in the list are peer-reviewed published conference proceedings. Items 5–7 are non-peer-reviewed annual reports published by NZ’s Department of Conservation. 

1. Brown K, Phillips, CB, Broome, K., Green, C., Toft, R, Walker, G. Feasibility of eradicating the large white butterfly (Pieris brassicae) from New Zealand: data gathering to inform decisions about the feasibility of eradication. Island invasives: scaling up to meet the challenge. Gland, Switzerland: International Union for the Conservation of Nature; 2019. pp. 364–369. Available: http://www.islandinvasives2017.com/

2. Phillips C, Brown K, Broome K, Green C, Walker G. Criteria to help evaluate and guide attempts to eradicate arthropod pests. IUCN Island invasives: scaling up to meet the challenge. Gland, Switzerland: International Union for the Conservation of Nature; 2019. pp. 400–404. Available: https://portals.iucn.org/library/node/48358

3. Richards N, Hardwick S, Toft R, Phillips C. Mass rearing Pteromalus puparum on Pieris rapae to assist eradication of Pieris brassicae from New Zealand. New Zealand Plant Protection. 2016;69: 126–132. 

4. Hiszczynska-Sawicka E, Phillips C. Mitochondrial cytochrome c oxidase subunit 1 sequence variation in New Zealand and overseas specimens of Pieris brassicae (Lepidoptera: Pieridae). New Zealand Plant Protection. 2014;67: 8–12.

5. Phillips C, Brown K, Green C, Walker G, Broome, K, Toft R, et al. Pieris brassicae (great white butterfly) eradication annual report 2013/14. Nelson, New Zealand; 2014 Dec p. 37. Available: http://www.doc.govt.nz/about-us/science-publications/conservation-publications/threats-and-impacts/animal-pests/pieris-brassicae-great-white-butterfly-eradication-annual-report/

6. Phillips C, Brown K, Green C, Broome K, Toft R, Shepherd M, et al. Pieris brassicae (great white butterfly) eradication annual report 2014/15. Nelson, New Zealand: Department of Conservation; 2015 Oct p. 19. Report No.: R77017. Available: http://www.doc.govt.nz/about-us/science-publications/conservation-publications/threats-and-impacts/animal-pests/pieris-brassicae-great-white-butterfly-eradication-annual-report/

7. Phillips C, Brown K, Green C, Broome K, Toft R, Shepherd M, et al. Pieris brassicae (great white butterfly) eradication annual report 2015/16. Nelson, New Zealand: Department of Conservation; 2016 Jul p. 20. Available: http://www.doc.govt.nz/about-us/science-publications/conservation-publications/threats-and-impacts/animal-pests/pieris-brassicae-great-white-butterfly-eradication-annual-report/

If published, our present manuscript (i.e. this one submitted to PLOS ONE) would be the first peer-reviewed article published in a journal rather than in conference proceedings about the NZ P. brassicae eradication program. Moreover, it would be the first/sole peer-reviewed article to provide an overview of the program (i.e. methods used and results obtained). The only other published work that provides a broadly similar overview of nearly the entire program is #7 in the above list, which is a non-peer-reviewed government annual report. However, this annual report was published before P. brassicae was officially declared eradicated from NZ and differs in many other ways to our PLOS ONE submission (one of the less obvious differences is that the data in our PLOS ONE manuscript are more correct than those in #7). The two earlier non-peer-reviewed government annual reports (#5 & #6) do not provide overviews because they were published before the eradication program ended (i.e. they are progress reports).

The peer-reviewed conference proceedings (#1–4 above) do not provide overviews of the program. Rather, they report very specific elements of it. We emphasise that contrary to the suggestion of Reviewer 2, the first item in the above list—a peer-reviewed conference proceeding—does not provide a comprehensive overview of the NZ P. brassicae eradication program. Instead, it describes uncertainties encountered when DOC and MPI were evaluating the technical feasibility and potential economic benefits of eradicating P. brassicae (i.e. before the program began). It differs substantially from our present manuscript and provides few details of how P. brassicae was eradicated.

Questions regarding financial disclosure

‘The authors have declared that no competing interests exist.'

We note that one or more of the authors are employed by a commercial company: Entecol Ltd

Authors: Updated financial disclosure

Details of all costs are publicly available on-line in the Department of Conservation’s 2015-16 Pieris brassicae eradication program annual report: www.doc.govt.nz/about-us/science-publications/conservation-publications/threats-and-impacts/animal-pests/pieris-brassicae-great-white-butterfly-eradication-annual-report/

Operational aspects of the eradication program were funded by the New Zealand Department of Conservation (DOC; www.doc.govt.nz). Vegetables New Zealand (www.freshvegetables.co.nz) contributed some funds to DOC to support operational aspects of the eradication program. DOC provided support in the form of salaries for authors K. Brown, CG and K. Broome, but did not have any additional role in the study design, data collection and analysis, decision to publish, or preparation of the manuscript. The specific roles of these authors are articulated in the ‘author contributions’ section. Vegetables New Zealand did not have any additional role in the study design, data collection and analysis, decision to publish, or preparation of the manuscript.

RT is a the Managing Director of a commercial company EntEcol Ltd (www.entecol.co.nz) which provides technical entomological services to New Zealand clients. In the eradication program, EntEcol Ltd was contracted by DOC for RT to provide services including contributing to the TAG, preparing documents, identifying specimens, helping to develop the visual lure, and evaluating P. brassicae parasitism rates. EntEcol Ltd provided support in the form of a salary for author RT, but did not have any additional role in the study design, data collection and analysis, decision to publish, or preparation of the manuscript. The specific roles of this author is articulated in the ‘author contributions’ section.

The New Zealand government research institutes AgResearch (www.agresearch.co.nz) and Plant and Food Research (www.plantandfood.co.nz) are partners in a New Zealand research collaboration called Better Border Biosecurity (www.b3nz.org). The collaboration aims to help reduce the rate at which non-native insects, weeds and diseases that could harm valued New Zealand plants are becoming established in New Zealand. AgResearch provided support in the form of a salary for author CP, but did not have any additional role in the study design, data collection and analysis, decision to publish, or preparation of the manuscript. The specific roles of this author is articulated in the ‘author contributions’ section. Plant and Food Research provided support in the form of a salary for author GW, but did not have any additional role in the study design, data collection and analysis, decision to publish, or preparation of the manuscript. The specific roles of this author is articulated in the ‘author contributions’ section.

The New Zealand Ministry for Primary Industries (www.mpi.govt.nz) provided financial support for some of the research costs of CP, GW and RT, and the New Zealand TR Ellet Agricultural Trust contributed support for some of the research costs of CP. MPI and TR Ellet Agricultural Trust did not have any additional role in the study design, data collection and analysis, decision to publish, or preparation of the manuscript.

Questions regarding Competing Interests section

Authors: Updated competing interests statement

The authors have declared that no competing interests exist. Author RT’s commercial affiliation to EntEcol Ltd does not alter our adherence to PLOS ONE policies on sharing data and materials.

Please include captions for your Supporting Information files at the end of your manuscript, and update any in-text citations to match accordingly. Please see our Supporting Information guidelines for more information: http://journals.plos.org/plosone/s/supporting-information

Authors: Captions added as requested.

Technical aspects

Editor’s comment

Also, I see that your figures show how the butterfly was eradicated, but I would like to see the quantitative impact of each particular action in time supported by statistical analyses. In this way, the efficiency and the timing of each action could be more objectively evaluated.

Authors: We were also motivated to do this during the eradication campaign to help us ascertain if and how P. brassicae was responding to each of the various treatments. Unfortunately, it was not, and still is not, possible to conduct such an analysis because: 

1. Different treatments were usually applied nearly simultaneously in either the same or proximate locations, thus their effects on P. brassicae were confounded.

2. It was too risky for the eradication program to leave some untreated areas as controls and the idea was never seriously considered (i.e. the eradication program was not a controlled experiment).

3. P. brassicae pupae were nearly impossible to detect and comprised an unknown and largely invisible proportion of the population for most of each year. This meant we were seldom able to obtain a reliable indication of the relative size of the whole P. brassicae population. The only time when few or no pupae were present in the population was in spring when all pupae—essentially the only P. brassicae stage that survived winter in Nelson—reached the adult stage and began reproducing: Adults, eggs and larvae were much more detectable than pupae. This nil-pupae period persisted through the latter part of September until mid October when the progeny of the spring adults began to reach the pupal stage. After spring, the P. brassicae generations became progressively less synchronised and some pupae were always present. Thus, we could only ever obtain reliable indications of the relative size of the whole P. brassicae population by comparing data between springs of different years. This single reliable annual estimate of relative population size was inadequate to separate the effects of different treatments, which had been applied together over the preceding year. (The autumn peaks in detection rates that are apparent in Fig. 3 arose for a different reason than the spring peaks, and were less useful for gauging the relative size of the total P. brassicae population because some pupae were still present during autumn, though fewer than in summer and winter.)

We note that where it was possible to roughly gauge the efficacy of certain treatments—in this case searching for eggs and larvae and applying insecticide—they are reported in the manuscript in the first paragraph of section ‘3.2 Active surveillance’. However, for the reasons previously described, it was not possible to estimate the population-level effects of these treatments.

Reviewers' comments:

3. Have the authors made all data underlying the findings in their manuscript fully available?

Reviewer #1: Yes

Reviewer #2: No

Authors: In the revision that we have provided an excel file of program data as supplementary information (S3 Data). We have aggregated these by management block to avoid providing details of individual properties.

5. Review Comments to the Author

Reviewer #1: 

One area I would like significant improvement is on pesticide applications - these are critical in eradication projects, but the deployment and use of this tool has not been covered adequately. Please respond to all queries in the "comments" call out on the attached PDF of this article.

L292 & L300- More details of insecticides requested.

Authors: Added a paragraph with the requested details to Methods section ‘Insecticides’.

L305- More details of herbicide requested.

Authors: Added a sentence with the requested details to Methods section ‘Host plant control’.

L317: Requested details of where parasitoids were reared.

Authors: Added a sentence with the requested details to Methods section ‘Biological control’.

L326: Requested details of method for feeding parasitoids.

Authors: Revised the sentence to say: ‘Cocoons were maintained until adult emergence, and adults were provided with 10% sugar solution via a vial with a cotton wick and allowed to mate’.

L355: Suggested change title from ‘Preparing this paper’ to ‘Data Analyses’.

Authors: Changed title to ‘Data presentation’ rather than ‘Data analysis’ as no formal statistical data analysis was conducted (or necessary).

L448: Define significant treatment.

Authors: Replaced ‘which also required significant treatment’ with ‘which were managed via 895 inspections of 314 properties’.

L479: More details of chemical treatments. Were they aerial?

Authors: Clarified in Results section ‘Host plant control’ that control was always from ground.

L486: Requested comment about impacts of natural enemies, biotic resistance.

Authors: As subsequently noted by the reviewer, this was covered in the Discussion in the paragraph that begins: “Several aspects of P. brassicae’s New Zealand habitat and ecology were fortuitously helpful to the program”. 

L558: Was P. brassicae ever considered an invasive threat to NZ?

Authors: Clarified in sentence added to 2nd paragraph of Introduction.

Additionally, there are a lot of "grey" references cited that appear to be almost irretrievable. I've highlighted a lot of these in pink in the attached PDF - I may have missed some, but I think I've selected sufficient to give the idea of the types of materials that I am concerned about. My feeling from reading the paper is that these "grey" documents have very useful information, hence their citation, and they need to be accessible for future use. Since they have been cited, the information is obviously not confidential. I think these materials need to be web accessible, either via supplemental materials associated with this paper, or via a "Giant White Butterfly Eradication Resource" website, possibly maintained by DOC(??!?!??)

Authors: DOC is solely focused on addressing critical conservation issues in NZ and does not have the resource to develop and maintain the suggested website. Our intention is to publish as much of the grey literature as possible to help make it widely accessible and the present manuscript is a key part of this effort. Another manuscript we are currently working on, for example, describes details of the P. brassicae phenology model that was developed to assist the eradication program. In the meantime, the authors of the present manuscript may also be able to provide unpublished reports to interested readers upon request.

Finally, perhaps in the discussion. I think it would be instructive to cover these three things: (1) the slow spread of the butterfly around Nelson - could this be due to a "lag" event - hence the critical 4 year window cited in the discussion? (2) The DNA work - can this be used to get a feel for founding population sizes and possible inbreeding that could've affected fitness (and subsequent rates of spread?), finally, Tobin/Liebhold/Suckling have made compelling arguments that you don't need to eradicate every last individual, you just need to get the population low enough so that destabilizing Allee effects can act and help push the population to extinction - do the authors have any thoughts on how Allee effects may have played a role in this program?

Authors: Our aim in this manuscript is primarily to describe how eradication was achieved in the hope that others will find it useful when considering and designing their own eradication attempts. We of course also wonder what role the processes mentioned by the reviewer played in the program’s success, but currently we are reluctant to increase the word-count with pure conjecture about those possible roles because we doubt it would help to evaluate and design future eradication attempts. It is possible that future analyses of specific aspects of the P. brassicae program might provide more quantitative, less speculative foundations for commenting about lag times, Allee effects etc. In the Discussion, we already describe processes that we have better reason to believe may have contributed to P. brassicae’s slow spread (see paragraph beginning ‘Throughout the program, doubt persisted that the feasibility criterion Immigration and emigration can be prevented’). And Hiszczynska-Sawicka & Phillips 2014 (cited in the manuscript and freely available on-line) used their P. brassicae genetic data to comment on possible founding population sizes etc and we think it is unnecessary to reiterate those comments in the present manuscript. Prompted by a suggestion of Reviewer 2, we have mentioned Allee effects (citing Tobin et al. 2011) in the revised final paragraph of the Discussion.

One last thing, biotic resistance is captured in the discussion, parasitoids/predators, no mention of pathogens (e.g., viruses/bacteria/fungi/microsporidia), any field observations to report on this guild of natural enemies? Any evidence for microsporidia in the invasive population that could have affected fitness??

Authors: Sentence about pathogens added to the paragraph that begins: “Several aspects of P. brassicae’s New Zealand habitat and ecology were fortuitously helpful to the program”.

Reviewer #2: 

The paper needs to more clearly acknowledge other published summaries of the same eradication program and explain the unique contribution that is being made herewith. For example, Brown et al. (2019; cited by the authors) provide a thorough overview of the eradication program and seem to summarize the same data. (It appears that small portions of the Brown paper are quoted verbatim or closely paraphrased without appropriate attribution; e.g. lines 73-75; 86-88.) Likewise, Figures 2 and 3 in the current paper seem remarkably similar/identical to figures in Phillips et al. (2016; cited by the authors).

Authors: The reviewer is incorrect that Brown et al. 2019 provides a thorough overview of the eradication program. Rather, it describes uncertainties encountered when DOC and MPI were evaluating the technical feasibility and potential economic benefits of eradicating P. brassicae (i.e. before the program began). It differs substantially from our present manuscript and provides very few details of how P. brassicae was eradicated. 

To avoid any possibility of self-plagiarism in the revised version of the present manuscript, we have changed the short sections of text noted by the reviewer and have also cited Brown et al. 2019.

We also note that Brown et al. 2019 is an article in peer-reviewed conference proceedings and, as requested, have uploaded it to PLOS ONE as supplementary information.

Phillips et al. (2016) is a non-peer-reviewed published Department of Conservation annual report, and Figures 2 and 3 in the current manuscript are similar, though not identical, to those in Phillips et al. (2016). If required, we will cite Phillips et al. (2016) in the captions for Figures 2 and 3, but currently we don’t believe this is necessary.

The true risk that P. brassicae posed/poses to native brassicas of New Zealand is not clear. It is not clear whether a formal pest risk assessment was ever completed for the species. Brown et al. (2019) note that the species was “[a]n Unwanted Organism under the Biosecurity Act of 1993, and a known pest of cultivated brassicas …”. The logic in the current paper seems to be that P. brassicae feeds on many species within the Brassicaceae; therefore, all Brassicaceae are at some risk of herbivory. Of course, this position is not supported empirically (i.e., oligophagous herbivores are not able to recognize and develop on ALL members of a plant family, Menken et al. 2010. Evolution 64:1098). Brown et al. (2019) provide some justification for the risk-averse position. However, a formal analysis could be done to compare the phylogenetic distance of plants of concern from known hosts. Such an analysis would be useful to justify the title of the paper, if it demonstrated that New Zealand natives were close evolutionarily to known hosts.

Authors: We agree with the reviewer and have made substantive changes to the Introduction to describe how many NZ native/endemic Brassicaceae belong to genera that contain host plants for P. brassicae in the Northern Hemisphere. See the two successive paragraphs in the Introduction starting with the one that begins: “New Zealand’s Department of Conservation (DOC) is responsible for protecting native biodiversity under the Conservation Act 1987”.

The authors highlight the fact that this effort appears to be the first attempted eradication of a butterfly. No explanation is given as to why this is significant. Is eradication of a butterfly expected to be significantly more/less difficult than the eradication of moths? Why?

We have added the following sentence to the 2nd paragraph of the Discussion: “The lack of powerful attractants for butterflies, which unlike moths use vision rather than long range sex pheromones to find mates (i Monteys et al., 2012), may have contributed to the dearth of previous attempts to eradicate butterflies (Kean et al., 2019).”

The methods begin with a significant discussion of the administrative structure of the eradication program. It is not clear which components of this structure should be considered essential. Many details seem better suited for supplementary details. More emphasis should be given to critical details, for example, a synopsis of the nine criteria and methods used to assess quantitatively the probability of eradication success. Phillips et al. (2019) review the criteria but do not describe how to use them to generate a probability of eradication success. Expert opinion appears to play a prominent role, but this is not specified.

Authors: We note that this Methods section describes the administrative structure rather than discusses it. We have shortened Methods section ‘2.1 Management and review’ and moved some of the less important details to ‘S1 Text Additional management details’. A subsequent comment of Reviewer 2 is “it would be helpful if the authors could comment on which aspects of the eradication effort would be most replicable in other situations”. Thus, in the revised Discussion (final paragraph), we briefly emphasise the importance of sound program management and its critical elements (further details are provided in response to this reviewer’s final comment).

Also in section ‘2.1 Management and review’, we have edited the paragraph that begins “The TAG developed nine criteria to help evaluate and guide the eradication attempt (Phillips et al., 2019)…” to clarify that the Phillips et al.’s (2019) criteria were not explicitly designed for estimating eradication probabilities. We then describe how TAG members were asked to informally derive their own probability estimates while using the criteria of Phillips et al. (2019) to help guide their thinking. We would prefer not to increase the manuscript’s word count with descriptions of the nine criteria, which are published and available on-line (https://portals.iucn.org/library/node/48358).

The authors do not provide an operational definition for ‘eradication’ at the outset. In the United States, three insect generations with intensive trapping and no catches are required before eradication can be declared. It is not clear whether this standard was met in New Zealand. The non-zero error bar at the end of the program in the current Fig 2 suggests that at least one P. brassicae might have been detected.

Authors: We have added the program’s initial operational definition of eradication to the end of the 2nd to last paragraph of the Introduction, then described at the end of the 1st paragraph of the Discussion how the decision was reached to declare eradication successful. 

As stated several times in the manuscript, the last detection occurred on 16 Dec 2014 and there were zero detections thereafter. In fact in Fig 2 every error bar is non-zero, but once detections reached zero they were usually too small to see on the graph. The few error bars that are visible during the period of zero detections appear simply because the sample sizes in those months were relatively small. During the 16 consecutive months of sampling when there were zero detections (Fig 2), the monthly sample sizes—i.e. properties inspected—ranged from 529 to 12250 with a mean of 6520 (S2 Data).

The structure of the eradication program lacks clarity. I consulted the Additional Information 2, which provides a map of the 46 management blocks. How much potentially suitable habitat for P. brassicae occurs within each block, how many total properties have suitable habitat within each block, and how many properties within each block were sampled? These are all critical details. The authors do not describe how data were analyzed. As the inspections do not appear to represent a simple random sample, the simple proportion of inspected properties that were infested is not an appropriate response variable. From the description, it appears that monitoring followed a stratified sampling scheme and resulting data should be analyzed accordingly.

Authors: In the final paragraph of the Results section ‘Active surveillance’, we have provided the requested details about potentially suitable habitat in each block, total properties with suitable habitat in each block, and how many properties in each block were sampled. We have also provided the data as a supplement in ‘S2 Data (an excel file). In particular, see worksheet ‘cadastre_and_search_data_by_block’, though we provide host plant data in the other worksheets too.

We have also updated the map of the P. brassicae management blocks (provided in the original version as Supp. Info. 2) to add some missing block names (S2 Figure Management blocks).

The reviewer is incorrect that the program followed a stratified sampling scheme because it, rather than stratifying properties, it repeatedly sampled the entire population of properties that had potential to be infested by P. brassicae. We have clarified this in the 1st and 2nd paragraphs of the Methods section ‘Prioritising locations to search’. 

We disagree with the reviewer’s suggestion that the proportion infested is an inappropriate response variable and the data should be analysed differently because: (i) As mentioned, the sampling was not stratified; (ii) The data were not statistically analysed and there is no response variable. Rather detection rates (infested properties/inspected properties) as used in the manuscript are a useful way of summarising and presenting the raw data. We consulted a specialist statistician (C. Van Koten, AgResearch) about the reviewer’s suggestions and she agrees with this response.

Given that insecticide treatments were highly-localized to individual host plants, it seems likely that some portion of the population would go untreated in each generation. Other eradication programs have relied on area-wide insecticide applications to achieve success. What proportion of the P. brassicae population would need to be treated in each generation to drive the population to zero in less than two years? Was such a level achievable? Could other factors (e.g., parasitism) have contributed to the success of the eradication?

Authors: We agree that unknown portions of certain P. brassicae life stages would have evaded treatment within each generation. This would have applied particularly to P. brassicae pupae, which were extremely difficult to detect and treat; even if area-wide insecticide applications had been applied, pupae would probably have remained impervious. We did attempt to augment parasitism of pupae, but did not measure any effect of the augmentation (Richards et al. 2016, cited in the manuscript). However, the program sought to treat every P. brassicae individual at some point in its development to force the population to decline between generations. It did this by targeting treatments at every life stage, though with less effort expended on pupae for reasons already mentioned. 

We do not have the data required—and nor did the eradication program have the resources to obtain them—to make a straightforward estimate of the proportion of each generation that must have been killed by the program to eliminate P. brassicae from Nelson in 2 years. We agree it would be interesting to model the eradication program by sampling variable values from assumed distributions for factors such as P. brassicae mating success, host finding, fecundity, and mortality due to parasitism, predation, disease, weather and the eradication treatments. However, such modelling has not been conducted, was not part of the eradication program, did not contribute to its success, and we think has no place in the current manuscript.

The reviewer asked if the (currently undefinable) required level of per generation mortality was achievable, and the answer is clearly yes because the population did decline to zero in about 2 years. The exact extent to which parasitism contributed to this decline is unknown, but we report notable levels of parasitism of larvae in the Results section ‘Biological control’ and speculate in the Discussion that natural enemies including parasites probably assisted population suppression (see paragraph that starts: ‘Several aspects of P. brassicae’s New Zealand habitat and ecology were fortuitously helpful to the program’).

In the end, it would be helpful if the authors could comment on which aspects of the eradication effort would be most replicable in other situations.

Authors: We agree and have revised the final paragraph of the Discussion accordingly.

---

## [Decision Letter · Decision Letter 1]

26 Jun 2020

PONE-D-19-32860R1

Eradicating large white butterfly from New Zealand eliminates a threat to endemic Brassicaceae

PLOS ONE

Dear Dr. Phillips,

Thank you for submitting your manuscript to PLOS ONE. After careful consideration, we feel that it has merit but does not fully meet PLOS ONE’s publication criteria as it currently stands. Therefore, we invite you to submit a revised version of the manuscript that addresses the points raised during the review process.

As in the previous round this article received two very different recommendations, the manuscript has been reviewed by one previous reviewer and two new ones in this second round. All the reviewers think the authors have addressed adequately previous comments, and that the work is very valuable, even though it not the typical research paper. However, the two new reviewers think there is still place for further improvements. Reviewer  I has several suggestions to improve the discussion and thinks the keywords chosen are not the most adequate ones. Reviewer II suggests adding some recommendations regarding the need of  indicators that can  be used in eradication campaigns, and the assessment of benefits of each specific eradication measurement. I concur with these reviews and I hope to receive a revised version of this manuscript that takes into account the suggestions raised.

We look forward to receiving your revised manuscript.

Kind regards,

Amparo Lázaro, PhD

Academic Editor

PLOS ONE

Reviewers' comments:

Reviewer's Responses to Questions

**Comments to the Author**

1. If the authors have adequately addressed your comments raised in a previous round of review and you feel that this manuscript is now acceptable for publication, you may indicate that here to bypass the “Comments to the Author” section, enter your conflict of interest statement in the “Confidential to Editor” section, and submit your "Accept" recommendation.

Reviewer #1: All comments have been addressed

Reviewer #3: (No Response)

Reviewer #4: All comments have been addressed

2. Is the manuscript technically sound, and do the data support the conclusions?

Reviewer #1: Yes

Reviewer #3: Yes

Reviewer #4: Yes

3. Has the statistical analysis been performed appropriately and rigorously? 

Reviewer #1: N/A

Reviewer #3: Yes

Reviewer #4: N/A

4. Have the authors made all data underlying the findings in their manuscript fully available?

Reviewer #1: Yes

Reviewer #3: Yes

Reviewer #4: Yes

5. Is the manuscript presented in an intelligible fashion and written in standard English?

Reviewer #1: Yes

Reviewer #3: Yes

Reviewer #4: Yes

6. Review Comments to the Author

Reviewer #1: This is an excellent piece of work that makes a significant contribution to the literature on eradication of invasive insect pests.

I would like to make a comment about the need for a "control" treatment - in this situation (i.e., eradication), as with area-wide management programs, a control treatment is not feasible, because of the risks outlined by the authors in their response letter.

Further, eradication programs are massive perturbation experiments, the system is studied prior to the implementation of the treatment, the system is "shocked" catastrophically (i.e., eradication tools are unleashed [insecticides, host plant eradication, manual collection and killing of pest insects]) and the system is evaluated during and after the "shock" is applied (climate change is a massive perturbation for which there is no control treatment but the data support overwhelming that humans are causing this event).

Outcomes, such as the elimination of the pest are almost certainly due to the perturbation and not due to some chance stochastic event.

The authors have thoroughly addressed the reviewer's comments.

The manuscript reads very well, it is very interesting and informative and will help with future eradication efforts, and it will also be cited a lot in the eradication literature. I found the "social science" and public engagement aspects of this work very interesting and helpful.

I recommend publication without any additional revisions.

Reviewer #3: Review of PLoS ONE Ms. PONE-D-19-32860R1 “Eradicating large white butterfly from New Zealand eliminates a threat to endemic Brassicaceae” by C.B. Phillips, K. Brown, C. Green, R. Toft, G. Walker and K. Broome.

General. This manuscript does not represent an “orthodox” publication. That is, one of experimental nature, hypothesis-driven, with extensive statistical analyses and in-depth interpretation/discussion of results. It is rather a thorough and very well written report of a noteworthy (“the first species of butterfly ever to have been eradicated worldwide”) and successful eradication effort of an invading, exotic butterfly (Pieris brassicae) that if established, would have put the local flora (particularly plants within the Brassicaceae) at risk in New Zealand. Having said the above, I believe the paper fits well into PLoS One.

I note that I have reviewed a version that underwent substantial changes after receiving many criticisms and suggestions for improvement (10 pages in small font) during the first round of revisions by two anonymous referees. In my opinion, all these criticisms have been thoroughly and completely/adequately addressed by the authors. The one most worrisome criticism pertaining to “self-plagiarism” and unwarranted reproduction of previously reported data/information, was, in my opinion, adequately dealt with. I have no additional concern about this issue in the current form of the manuscript.

Despite the above, I believe there is still room for additional improvements. I refer particularly to the Discussion that needs to be compacted and start with an initial paragraph summarizing the main findings and pointing to the principal issues that will be discussed in what follows. In its current form, the authors jump straight into discussing details. I also believe that even though Pieris brassicae is a broadly known species, the expert taxonomist that confirmed the identity of the specimens needs to be acknowledged, the place where voucher specimens are housed needs to be specified, and further discussion on the possible origin on the invading “founders” based on the genetic studies performed is needed.

Further details in the next section.

Specific.

L. 1 – Tittle. Is not a “the” missing after “Eradicating”. That is, “Eradicating the large white butterfly …..

L. 34 – Abstract. Add “worldwide” in the end. That is, “This is the first species of butterfly ever to have been eradicated worldwide”.

L. 36-37. I do not believe that the keywords chosen by the authors are the ideal ones. In my opinion, the following words would be much better suited: eradication, Pieris brassicae, Lepidoptera, New Zealand, invasion biology, biosecurity, endemic flora conservation.

L. 76 – Introduction. Here and many places elsewhere in manuscript (e.g., lines 305, 472, 483, 710, 726) numbers are used that should be spelled out in words. The grammatical rule indicates that any number smaller that 10 needs to be written in words not numbers. Here (L. 76) it reads “….. to Nelson 2 years after ….”. Believe it should be “two years”. Leave this to the editor as I do not know the specific rules of PLoS One.

L. 118 – Methods. It would be very useful for the reader to have access to a summary figure with a clear flow chart detailing the most important strategies followed during the entire eradication program from start to end. A lot of actions were implemented, and the reader can get overwhelmed with so much detail. Therefore, a tidy, well thought out/designed flow chart leading the reader step by step as to the critical strategies followed in the “command room” by the chief program manager, would greatly simplify reading/perusing over all the details.

Somewhere in this section, the authors need to specify who formally/taxonomically identified Pieris brassicae, Cotesia glomerata, Pteromalus puparum, indicate in which officially recognized insect collection are the voucher specimens saved, and also where is the genetic material saved for future verifications (i.e., mitochondrial COI gene sequences) in the case of Pieris brassicae. Independent of the fact that P. brassicae is a well-known insect, these types of formalities cannot be overridden.

L. 435/436. Explain if such a broad range (32-52% and 5-25%) did not entail the risk of undetected eggs/larvae. This is a minor point as the end result was eradication …

L. 466. Capitalize N in “nasturtium“.

L. 476-479. A bit confusing. Was every telephone report (direct or toll-free) later physically confirmed by an expert? Was the ID of all specimens verified?

L. 481-485. Any idea as to what the origin of the invading population was? The authors mention that they kept mitochondrial COI gene information on all specimens sampled. If so, this information could be matched with data kept at official repositories (e.g., gene bank) and obtain an idea with respect of the origin of the invaders. This information would be useful to avoid reinvasions, and to strengthen pathway regulations.

L. 563-605 are repeated in Lines 607-48.

L. 563 – Discussion. This section needs to be started with a paragraph summarizing the most significant results of the previous section (Results), outlining in the end the approach that will be followed in discussing these relevant findings. Currently, the first paragraph of the Discussion (Lines 564-576), leads the reader too abruptly into details that do not provide an overview of the relevant findings.

In addition, I feel the entire Discussion section could be tightened/compacted and better organized.

Reviewer #4: Comments to the editor and corresponding Author

Since this is a second round review, I first went through the previous reviewers’ questions and the authors’ responses as well as the subsequent edit they made to the manuscript in response to the reviewers’ concerns. I would like to congratulate the previous reviewers on the critical and robust interrogation they made on the original manuscript. I am also intrigued by the detail at which the authors responded to each and every question. Even if it is quite difficult to judge on behalf of the original reviewers’ as they probably have a set way they expected concerns to be addressed, from my point of view I believe that the concerns raised are fully addressed.

I would here like to comment that I understand the reservations put forward by one of the previous reviewer that quantitative assessment of the measures taken to eradicate Pieris brassicae is lacking. However, I believe it is important to recognize that such eradication efforts are usually run by mandated government organizations whose primary goal is to eradicate the species; therefore, any activity that might be really interesting or important from future research point of view might not necessarily be prioritized. Yet, since the eradication was a success even if it is hard to quantify which specific measure contributed how much towards the success, it is still a very important source where future researchers and managers alike can find detailed prescriptions of how each measure was implemented. And since the sum of these different measures, save the natural enemies and terrain, resulted in a successful eradication the article already would make one tried and tested integrated eradication management system available for end-users.

I have the following two recommendations for the authors

Recommendation 1

Mandated bodies usually take recommendations from researchers to optimize the eradication but not necessarily take steps to provide data that can be used to assess the efficacy of taken measure with the same rigor they carry out the eradication itself. In this regard, I believe it is appropriate for the authors to insert a recommendation for biostatisticians/pest risk modelers/population researchers/entomologists or others in a similar field to come up with a new set of - or modified indicators or parameters that can easily be measured by eradication campaigns. This will help quantitatively assessing the exercise without taxing the body who is carrying out the eradication too much time and effort for collecting the necessary data.

Recommendation 2 ( this one is if the authors feel inclined and if they think it is not out of the scope of their paper)

As I mentioned a couple of times above, there is not enough information on details of how each of the measures taken contribute towards the final success. It is usually very difficult to attain such detail as such eradications are often carried out under limited resources and under government oversight making it difficult, to collect and analyze data needed for such analysis.

In short, it is not run like an experiment rather it is a job that needs to be done. However, I believe there is still a case to be made to such bodies that the more of such information they collect even if costly, it will in the end make overall eradication efforts economically sound. Therefore, It will be beneficial for future eradication endeavors if the authors recommend in their paper that MPI/DoC or any mandated body that might carry out such eradications to make room in their program to assess the benefits of each specific measure within their integrated eradication system. Such data is elemental in building bio-economic eradication models that can optimize success both in the economic and biological sense. Although, in such cases when endangered species are involved money might not be an issue, overall if there is data that can optimize eradication techniques and drive down cost. The decision to manage rather than eradicate harmful invasive species might have a different outcome.

Line by line (minor)

L55: I believe in some ways the good outcome of the eradication effort is in some implicit way helped by the good system-wide understanding of best biosecurity practices in NZ. By that, I mean the increased awareness regarding biosecurity in New Zealand among the Government, Researchers and Citizens alike. It is important to give some reference besides the P. brassicae being listed as Unwanted Organism, that it has also been identified as a possible threat and invader by New Zealander researchers. For example Worner and Gevrey (2006) predicted its possible invasion in 2006.

L551: the use of a colon here rather signaled a list but what followed was two items separated by a semi colon. Better to remove the colon and continue with “allowed…” and replace the semi colon with a comma on L552. It will make it easy to read the paragraph.

Reference: Worner, S., & Gevrey, M. (2006). Modelling global insect pest species assemblages to determine risk of invasion. Journal of Applied Ecology, 43(5), 858-867.

7. PLOS authors have the option to publish the peer review history of their article (what does this mean?). If published, this will include your full peer review and any attached files.

Reviewer #1: No

Reviewer #3: No

Reviewer #4: **Yes: **Senait D. Senay

---

## [Author Response · Author response to Decision Letter 1]

9 Jul 2020

We have revised the manuscript in response to the editor's and reviewers' recommendations. We do not have laboratory protocols to deposit in protocols.io.

---

## [Editor Report · Decision Letter 2]

15 Jul 2020

Eradicating the large white butterfly from New Zealand eliminates a threat to endemic Brassicaceae

PONE-D-19-32860R2

Dear Dr. Phillips,

We’re pleased to inform you that your manuscript has been judged scientifically suitable for publication and will be formally accepted for publication once it meets all outstanding technical requirements.

Kind regards,

Amparo Lázaro, PhD

Academic Editor

PLOS ONE
---

## [Editor Report · Acceptance letter]

17 Jul 2020

PONE-D-19-32860R2 

Eradicating the large white butterfly from New Zealand eliminates a threat to endemic Brassicaceae 

Dear Dr. Phillips:

I'm pleased to inform you that your manuscript has been deemed suitable for publication in PLOS ONE. Congratulations! Your manuscript is now with our production department. 

Kind regards, 

on behalf of

Dr. Amparo Lázaro 

Academic Editor

PLOS ONE